# DiffDock-Pocket: Diffusion for Pocket-Level Docking with Sidechain Flexibility

## Abstract

When a small molecule binds to a protein, the 3D structure of the protein and its ability and efficiency to perform its function can significantly change. Understanding this process, called *molecular docking*, can be crucial in areas such as drug design. Recent learning-based attempts have shown promising results at this task, yet lack features that traditional approaches support. In this work, we close this gap by proposing DiffDock-Pocket, a diffusion-based docking algorithm that is conditioned on a binding target to predict ligand poses only in a specific binding pocket. On top of this, our model supports receptor flexibility and predicts the position of sidechains close to the binding site. Empirically, we improve the state-of-the-art in site-specific-docking on the PDBBind benchmark. Especially when using *in-silico* generated structures, we achieve more than twice the performance of current methods while being more than 20 times faster than other flexible approaches. Although the model was not trained for cross-docking to different structures, it yields competitive results in this task.

## 1 Introduction

Proteins are the building blocks of life and are ubiquitous in biochemical processes of all organisms. They realize various biological functions by interacting with other biomolecules, such as other proteins or small ligands. The 3D structure of each protein governs the possible interaction partners, which can play a crucial role in their function. When a molecule (ligand) interacts with a protein (receptor) and binds to it, they form a so-called complex that can have a different function [Stank et al., 2016]. Accurately predicting these molecular interactions can give insight into the inner workings of biological processes and is thus a highly important task in computational biology and drug discovery [Kubinyi, 2006; Meng et al., 2011; Pinzi & Rastelli, 2019]. Molecular docking aims to predict these interactions by determining the 3D position of the ligand when bound to the receptor.

In drug discovery campaigns, the processes underlying diseases are usually well-researched and specific targets can often be identified, which, if modified or inhibited, can potentially treat a disease [Weisel et al., 2009]. This means a specific part of the protein (e.g., a druggable pocket) is often known to be responsible for a biochemical interaction and is thus the target of a docking procedure [Zheng et al., 2012]. Site-specific docking incorporates prior knowledge of a binding site and limits possible docking poses of a given ligand to a specific receptor region. This reduces the search space by a large margin, simplifying the docking problem. Many machine-learning (ML) based approaches cannot account for prior knowledge of a pocket [Stärk et al., 2022; Lu et al., 2022; Corso et al., 2023], despite the need in practical applications for docking to a specific target. This is seen as one of the most significant limitations of current ML approaches [Yu et al., 2023].

Therefore, we consider the task of pocket-level docking and additionally model receptor flexibility of the sidechain atoms near the binding site. When a ligand docks to a receptor, they both undergo conformational changes [Huang, 2017], with the sidechain atoms in the binding site displaying the most significant ones [Clark et al., 2019]. Understanding and modeling sidechain flexibility is critical in molecular docking [Teague, 2003], as it can directly influence the prediction accuracy [Zhao & Sanner, 2007; Hogues et al., 2018]. Many current methods either ignore this issue and model rigid receptors [Stärk et al., 2022; Lu et al., 2022; Corso et al., 2023], or adding flexibility significantly impacts the accuracy and runtime [Koes et al., 2013b; McNutt et al., 2021], making them unsuitable for large-scale tasks such as screening drug candidates. We believe that fast, accessible, and reliable

DIFFDOCK-POCKET

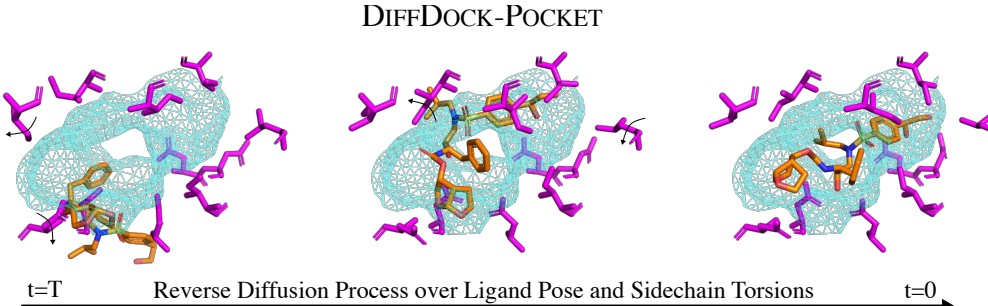

t=T          Reverse Diffusion Process over Ligand Pose and Sidechain Torsions          t=0

Figure 1: **Overview of our approach.** The model takes as an input a ligand, an (*in-silico* generated) protein structure, and the binding target. The process starts with random ligand poses (orange) and sidechain conformations (magenta), which are gradually improved by a reverse diffusion process (left to right) to represent meaningful results. The generative process modifies the translation, rotation, and torsional angles of the ligand and the torsional angles of the receptor's sidechain atoms to predict a final pose for each. This is all done with the knowledge of a binding pocket (blue).

site-specific docking with flexibility can drive discovery in computational biology, especially in drug design.

This paper takes a step towards solving this problem by proposing DIFFDOCK-POCKET: a diffusion-based model for pocket-level molecular docking with receptor sidechain flexibility inspired by the ideas of DIFFDOCK [Corso et al., 2023]. It uses diffusion over a reduced product space to predict sidechain and ligand confirmations, as illustrated in Figure 1. Moreover, our approach narrows the performance gap when docking to *in-silico* generated structures, which, while not exact, often provide strong approximations and are readily accessible.

Our model demonstrates state-of-the-art performance in the PDBBind [Liu et al., 2017] docking benchmark, where we achieve a root mean squared deviation (RMSD) of less than 2Å in 49.8% of cases compared to 27.8% achieved by the best method evaluated with receptor flexibility. All other tested approaches suffered majorly in terms of accuracy and runtime when modeling the receptor as flexible (DIFFDOCK-POCKET is 25–90 times faster than other flexible approaches). When relying on *in-silico* generated protein structures, the model retains most of its capabilities for docking and sidechain predictions. We achieve scores of 41.7% and 39.5% for *in-silico* structures generated from ESMFold2 [Lin et al., 2022] and ColabFold [Mirdita et al., 2022] respectively. On the CrossDocked 2020 benchmark [Francoeur et al., 2020], our model yields better pocket-normalized docking scores than other methods, despite some of the other approaches being specifically trained on this dataset.

Our main contributions can be summarized as follows:

1. We introduce a diffusion model for site-specific docking with receptor flexibility, yielding better results than all freely available approaches on the PDBBind benchmark.

2. We show that by including computationally generated structures in the training procedure, our model can retain a high performance when using *in-silico* generated structures.

3. We demonstrate competitive cross-docking performance by evaluating it on a subset of the CrossDocked 2020 dataset, with proteins removed that were seen during training.

## 2  RELATED WORK

**Molecular docking.** Docking a small molecule to a protein is a complicated biochemical process governed by the energy of the interacting atoms. During docking, the protein and ligand atoms orient themselves and take on the conformation that results in the most energetically favorable binding configuration. Using this knowledge, traditional search-based models such as GLIDE, [Friesner et al., 2004; Halgren et al., 2004], MOLDOCK [Thomsen & Christensen, 2006], and AUTODOCK [Trott & Olson, 2010] minimize a scoring function that calculates the energy of a given configuration (based on the force fields or statistical potential recovered from experimental data). Approaches such as GNINA [McNutt et al., 2021] and DEEPDOCK [Méndez-Lucio et al., 2021] use ML to approximate this score function, while others such as SMINA [Koes et al., 2013b] take a more classical ap-

proach. SMINA and GNINA are considered among the best freely available search-based docking solutions, whereas GLIDE enjoys commercial success.

Minimizing the scoring function over the whole search space can be challenging. However, since key binding regions are often already known through experimental data, the search space can be limited. Most approaches, especially classical ones, can typically limit the search space to this pocket rather easily. ML based approaches such as DIFFDOCK [Corso et al., 2023], EQUIBIND [Stärk et al., 2022], and TANKBIND [Lu et al., 2022] usually fail to account for binding pockets completely.

**Flexible docking.** Almost all recent docking approaches model the ligand flexible [Huang, 2017; Koes et al., 2013b; McNutt et al., 2021], but some do not account for the changes that can occur in the protein [Friesner et al., 2004; Halgren et al., 2004; Stärk et al., 2022; Lu et al., 2022; Corso et al., 2023]. These geometrical changes can play a crucial role in successfully modeling a binding process because already slightly different receptor conformations can change the energetically optimal structure. Algorithms that fail to account for receptor flexibility can lose accuracy because wrong configurations can make correct docking positions biochemically unlikely [Zhao & Sanner, 2007; Hogues et al., 2018]. Modeling flexibility is thus especially important when no re-docking is performed (i.e., docking to a ligand-free structure) and realistic sidechain positions are unknown.

Since predicting the position of each atom of a protein is a computationally expensive task, especially for large proteins, most algorithms used in practice nowadays model the proteins semi-flexible [Meng et al., 2011]. This method is motivated by the chemical properties of the protein because peptide bonds between the amino and carboxyl groups of the amino acids are rigid in nature, so the protein's backbone is usually rigid. However, the parts of the amino acids that extend outwards from the $\alpha$-carbon atom (i.e., the sidechain atoms) display more flexibility and undergo the majority of structural changes, especially near the binding site [Clark et al., 2019].

Search-based approaches such as GNINA or SMINA can include the additional sidechain atoms in their stochastic energy-optimization procedure. However, this can drastically increase the search space, and the computational effort and thus reduce the accuracy. For ML models, modeling receptor flexibility can be challenging and is typically unsupported [Corso et al., 2023; Stärk et al., 2022; Lu et al., 2022]. NEURALPLEXER [Qiao et al., 2023], is a recent diffusion-based docking algorithm that can predict all atom coordinates of the protein and the ligand within a specified pocket by masking the target and predicting new coordinates. However, as of writing, no code is available.

**Diffusion.** Previous work [Corso et al., 2023] has shown that generative modeling is well-suited for docking due to its ability to capture the stochastic nature of the biological process and its uncertainty. Score-based diffusion models [Song et al., 2021] define a continuous diffusion process $dx = f(x,t)\,dt + g(t)\,dw$ to apply to points of the data. Critically, this has a corresponding reverse SDE $dx = [f(x,t) - g(t)^2 \nabla_x \log p_t(x)]dt + g(t)\,dw$ where only the score $\nabla_x \log p_t(x)$ is unknown. Throughout this paper, $f(x,t)$ will be 0. Given an initial distribution $p_0$ (the distribution of the data), if the evolving score is learned, the reverse equation can be numerically solved to produce new points of the underlying data distribution from random noise. For molecular docking, this means that beginning from a random starting conformation of the ligand, noise can be removed such that the end conformation will be the state of the ligand docked to the target protein.

## 3 METHOD

Given a ligand and a protein, flexible docking models predict the geometrical structure of both the ligand and the protein. Assuming a fixed scaffold, the structure of this binding complex is uniquely described by its atom positions in the three-dimensional space. For a ligand with $n$ atoms, and a protein with $m$ flexible atoms, the space of possible predictions is in $\mathbb{R}^{3(m+n)}$. The large space w.r.t. the number of data points available makes docking a challenging problem. Especially for large proteins with thousands of atoms, searching for an optimal conformation of all positions is computationally infeasible.

The first step we take is to make the search space smaller by reducing its dimension using knowledge about the rigidity of different molecular transformations. Instead of modeling the protein and ligand with all their 3D atom coordinates, the conformations can also be described by the changes the ligand and the sidechains undergo during binding. The main biochemically possible changes are the rigid 3D translation or rotation of the complete ligand w.r.t. the receptor and the rotation of the torsion

angles of the ligand's chemical bonds. Similarly, the backbone of the receptor stays mostly rigid, and mostly the torsional angles of the receptor sidechain atoms change. These transformations form an algebraic group structure and together span a $3 + 3 + k + \ell$ dimensional manifold, which we refer to as the *product space*. $k, \ell$ are the number of torsion angles in the ligand and protein respectively. While this does not cover all possible conformations of the protein and ligand, it accounts for the most prominent changes and keeps properties such as the rather stable bond lengths fixed. By applying the knowledge of possible modifications and searching in the product space, we reduce the dimensionality of the search (see Appendix A), excluding chemically unlikely structural changes. This way, we can aim to learn the scores on the tangent spaces of the transformation manifold and only predict these four lower-dimensional changes to the initial structure.

## 3.1 SITE-SPECIFIC DOCKING

Since docking sites are often known or chosen in advance, we can further reduce the space and speed up the search for an optimal conformation by including this prior information. With this, we can expect more accurate results while requiring less computational effort. Various ways exist to condition the model to a known binding pocket, depending on the underlying method used. Diffusion models build on the idea that they iteratively refine a random initial configuration. To condition the ligand pose on a binding pocket, we propose to center the ligand's initial random configuration around the pocket's center while also limiting the maximum translation our model can predict. With this change, all ligand poses are guaranteed to be within the target pocket, but the model still needs to predict a (small) translation to account for the random noise and different poses. Formally, the random ligand translation $z_{tr}$ will be sampled from a normal distribution with a relatively small variance. This will have no effect on the initially random rotation and torsion angles.

However, for large proteins, this would still mean that our approach needs to consider atoms far away, although the atoms close to the binding site influence the actual binding procedure most. By exploiting this fact, we decided to discard all amino acids that are too far away from the target binding site, as depicted in Figure 2. This focuses the model's attention on the binding site and reduces all proteins to a similar size. Additionally, this reduced view of the protein allows us to represent even large proteins using only a comparatively small subset of amino acids and did not lead to a decrease in accuracy in our tests. With this, all atom positions can be used as input to the model instead of simply using the coordinates of the backbone (C-$\alpha$ atoms), as was done in previous work [Corso et al., 2023]. This allows our model to learn more physics-informed predictions, potentially improving the accuracy.

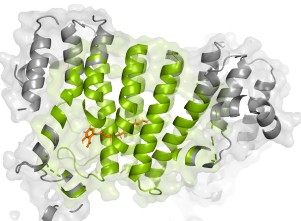

Figure 2: **Pocket reduction.** Only retain amino acids close to the ligand (green) and discard all others (gray).

We require knowledge of the pocket center position in $\mathbb{R}^3$ and a radius indicating the pocket's size to center the translational noise and reduce the protein. As for the pocket size, we use the radius of the smallest sphere centered at the mean of the ligand that can fit all atoms. We then also add an additional buffer of 10Å to the radius to retain the surrounding context of the pocket for the model to make predictions. If any atom of an amino acid falls within this distance from the pocket center, the whole amino acid is kept, whereas all other amino acids are discarded. Defining the pocket center can be a bit more challenging because, in practice, one might be able to infer the general area where a ligand might dock but cannot pinpoint the exact center of the ligand. To avoid bias in the training data, we calculate the pocket center by taking the average positions of the C-$\alpha$ atoms within 5Å of any ground truth ligand atom. This technique aligns with a setting where one would visually analyze the protein and suspect the pocket location. By only using the rigid backbone to calculate the center, this definition of a pocket works well, even when the protein has a different sidechain structure.

## 3.2 FLEXIBLE SIDECHAINS

In principle, any of the remaining amino acids can be modeled flexibly. However, implementing flexibility for all residues would again increase computational complexity (although manageable with this reduced product space) without providing much benefit as it has been shown that flexibility is mostly restricted to the residues close to the binding site [Clark et al., 2019]. Therefore, we follow

the convention from other docking algorithms [McNutt et al., 2021], and model only amino acids which have at least one atom within 3.5Å of any ligand atom as flexible.

Once the flexible sidechains have been selected, the concrete rotatable bonds have to be determined. A graph is constructed for each residue based on the chemical order of atoms inside the sidechain. Each connecting edge then describes one rotatable bond (refer to Section C.3). This way, the conformation of the sidechains can be approximately described by the torsion angles of each rotatable bond, and the model can learn to predict the score of these angles. Formally this means that depending on the concrete amino acid $a$, the model predicts $\ell^a$ ordered torsion angles $\mathcal{X}_1^a, \ldots, \mathcal{X}_\ell^a$. Rotating the torsion angles of each sidechain bond of the protein $\boldsymbol{y}$ by the predicted angles $\mathcal{X}$ yields the new atom positions $\tilde{\boldsymbol{y}}$. Although all angles $\mathcal{X}$ are predicted simultaneously at each timestep, they are iteratively refined by the diffusion process. This has the advantage that the angles can influence each other without sacrificing performance compared to doing it autoregressive.

## 3.3 SIDECHAIN CONFORMER MATCHING

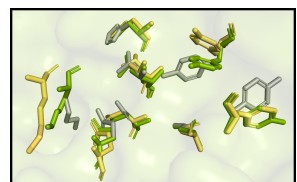

Figure 3: **Sidechain conformer matching.** Optimize the sidechain torsional angles (green) of the computationally generated structure (gray) to minimize the distance to the ground truth positions (yellow).

When learning the torsional angles with our proposed diffusion approach, we need access to a protein where the atoms of the flexible sidechains are bound to the ligand. Holo crystal structures already contain this correct information. A problem arises when we want to rely on different data for our method, such as either computationally generated structures or proteins that are bound to different ligands. In these cases, the torsional bond angles and bond lengths will be different from the ground truth data. This shift can be attributed to other (non-prominent) conformational changes the protein undergoes (e.g., the lengthening or shortening of bonds) or to inaccuracies of predictive models when using synthetic data (i.e. ESMFold is not perfect). Hence, such protein structures cannot be used directly for training.

To still be able to expose the model to different structures, we prepared computationally generated structures with a procedure referred to as *sidechain conformer matching*. The idea is to align the torsional angles of the computationally generated structures to the ground truth ligand-bound crystal structures while keeping the rigidity of the bonds, as can be seen in Figure 3. This allows us to use computationally generated structures for the training procedure while having sidechain positions that are close to the ground truth. At inference time no sidechain conformer matching will be performed, as this could leak test data.

Similarly to Jing et al. [2022], we define the search for these structures as a minimization problem of the RMSD between the ground truth structure $\boldsymbol{y}$ and *in-silico* structure $\boldsymbol{y}'$ over the torsional angles of the flexible amino acids. When referring to the ligand as $\boldsymbol{x}$ and assuming we have a sidechain for amino acid $a$ with $\ell^a$ rotatable bonds $\mathcal{X}_1^a, \ldots, \mathcal{X}_\ell^a$ the goal can be phrased as $\ell$ minimization problems for each amino acid

$$\text{match}(\boldsymbol{x}, \boldsymbol{y}, \boldsymbol{y}') = \underset{\tilde{\boldsymbol{y}} \in \{\text{apply}(\boldsymbol{y}', \mathcal{X})\}}{\arg\min} \text{RMSD}(\boldsymbol{y}, \tilde{\boldsymbol{y}}) \cdot \text{penalty}(\boldsymbol{x}, \tilde{\boldsymbol{y}}). \tag{1}$$

The additional penalty in the optimization goal was introduced to make the matched proteins more realistic. It aims to reduce the number of steric clashes (i.e., atoms that would be too close together), and is described in more detail in Appendix B. The minimization is solved with differential evolution, which iteratively combines potential solutions of a population to converge to the global minimum. We can then use the computationally generated structure where the sidechains have been conformer-matched with the bound structure in training. This matching still leaves some distance between the structures (as seen in Figure 3) but aligns with our definition of a semi-flexible receptor.

## 3.4 MODEL ARCHITECTURE AND TRAINING

**Models.** The model architecture we are using is inspired by the structure of DIFFDOCK [Corso et al., 2023] and consists of two different models which are executed in sequence during inference: the score model and the confidence model. The aim of the *score model* is to learn the (diffusion) scores of the tangent spaces of the transformation manifolds: $\mathbb{T}^3$ for translation, $SO(3)$ for rotation, $SO(2)^k$ and $SO(2)^\ell$ for the torsion angles of the ligand and flexible sidechains respectively.

With the knowledge of the scores during inference, we can take a protein with pocket and a ligand structure in 3D space and produce $i \in \mathbb{N}$ different complex structures $\left(\tilde{\boldsymbol{x}}^{(1)}, \tilde{\boldsymbol{y}}^{(1)}\right), \ldots, \left(\tilde{\boldsymbol{x}}^{(i)}, \tilde{\boldsymbol{y}}^{(i)}\right)$.

The *confidence model* is then used to rank each protein-ligand prediction such that the best-predicted structures can be selected. Our training routine and objective are defined so that our confidence model learns to predict the accuracy of generated binding structures by considering both the ligand's docking success and the similarity of flexible sidechains to the bound structure. The output of the confidence model is a logit and important for real-world application since it allows practitioners to judge the accuracy of the predictions without access to the ground truth.

**Architecture.** The architecture between both models is very similar and mostly differs in the last few layers. Since we are learning the distributions on the transformation space instead of the three-dimensional positions, we can formulate a desirable generalization of the model by exploiting attributes of group actions. Mainly, we want our model to recognize the similarity or equivalence of complex structures that can be transformed into each other using distance-maintaining ($SE(3)$) transformations. Therefore, we expect our output scores on the rotation and translation tangent spaces to be $SE(3)$-equivariant and our torsion angle scores to be $SE(3)$-invariant. We achieve this by using $SE(3)$-equivariant convolutional networks, so-called tensor field networks [Thomas et al., 2018; Geiger et al., 2022] that encode the data into irreducible representations of the $O(3)$ group.

In our architecture, both the ligand and protein are represented as geometric graphs where nodes represent atoms and edges are between close neighbors or chemical bonds. There are edges between ligand-ligand nodes, receptor-receptor nodes, and also receptor-ligand nodes. Moreover, we also define a graph for each amino acid in the receptor instead of every atom. This representation follows multiple convolutional layers, where we make use of message passing between the nodes based on the node and edge features. In the end, this yields representations for each atom.

After the convolutional layers, the architecture between the score and confidence model differ, as they have different objectives. The score model needs to output a translational score, a rotational score (around the center of the mass of the ligand), and one torsional score for each of the $k$ rotatable bonds of the ligand. To allow for a flexible receptor, the score model also needs to predict $\ell^a$ torsional scores, one for each rotatable bond in every flexible amino acid $a$. For this, we use a pseudotorque layer as introduced by [Jing et al., 2022] similar to the architecture predicting the torsion scores of the ligand. For the concrete diffusion process on torsional angles, we refer to [Jing et al., 2022; Corso et al., 2023]. As opposed to the score model, the confidence model is not diffusion-based and thus does not predict any scores. The output is a single $SE(3)$-invariant scalar, which is predicted by an MLP that uses the flexible atom and ligand representations. It uses the predicted structures as input and aims to determine the probability that the docking is accurate.

**Training.** We use diffusion score-matching [Song et al., 2021] to train our score model by sampling the transformations from the perturbation kernels, applying them to the input structures of our model, and minimizing the theoretical denoising score matching loss function for each transformation $T$

$$\boldsymbol{\theta}^* = \arg\min_{\boldsymbol{\theta}} \sum_{\text{trf} \in T} \mathbb{E}_t \left\{ \lambda(t) \mathbb{E}_{\mathbf{x}(0)} \mathbb{E}_{\mathbf{x}(t)|\mathbf{x}(0)} \left[ \left\| \mathbf{s}_{\boldsymbol{\theta}}^{\text{trf}}(\mathbf{x}(t), t) - \nabla_{\mathbf{x}(t)} \log p_{0t}^{\text{trf}}(\mathbf{x}(t) \mid \mathbf{x}(0)) \right\|_2^2 \right] \right\}, \quad (2)$$

as described in Song et al. [2021], with $\lambda(t)$ a positive weighting function for each time $t$. The minimization is done while iterating through the conditional distributions corresponding to each ligand-protein pair. This formulation is equivalent to minimizing the distance between the real and predicted scores of the conditional distribution.

To train the confidence model, we first sample diverse ligand and sidechain configurations with the score model. The predictions are then compared with the ground truth training data to assess their quality. The confidence model learns to predict this quality by training it with a binary cross-entropy loss on those generated structures to predict if the sampled configuration is plausible.

**Inference.** To predict a docked complex, we start from an arbitrary ligand and flexible sidechain conformations by applying random transformations in all degrees of freedom. We then use the score model to predict the transformation scores of the conditional distributions at each timestep and use the output to construct the reverse stochastic equation. Intuitively, by solving the reverse diffusion equation, we iteratively move the samples to regions with high densities of the underlying distribution by following the vector field produced by the predicted scores. Once the diffusion process is finished, the samples are ranked based on their quality estimated by the confidence model.

Due to the maximum likelihood training, the predictions of the score model can be dispersed over multiple modes of the target distribution. We perform low-temperature sampling to prevent this problem of overdispersion at inference due to model uncertainty and thereby emphasize the modes of the distribution. This is done via the approach proposed by Ingraham et al. [2022, Apx. B]. For this, we have determined the temperature values for our score model that maximize its performance on the validation set.

## 4  RESULTS

Obtaining real-world data in molecular biology can be challenging, and the limited available data must be used meaningfully. This can make it difficult for docking algorithms when the distribution of the structures changes. In this section, we will demonstrate that our model generalizes well beyond the data seen and exhibits high performance over different tasks, including docking to computationally generated structures and docking to proteins originally bound to a different ligand. We will also show that our model can be used to improve the sidechain configuration of *in-silico* generated protein structures to better account for the ligand-bound structure. The source code and documentation of our model is available at `https://anonymous.4open.science/r/DiffDock-Pocket-AQ32`, and the versatile interface allows it to be run with many different formats, pockets, and with any number of flexible amino acids.

**Setup.** As a training set, we relied on PDBBind [Liu et al., 2017], a subset of PDB [Berman et al., 2003], with a time-based split and a mixture of crystal and ESMFold2 generated structures. In this section, we evaluate it on the unseen testset. We either used the crystal structure from PDBBind or computationally generated structures with the same amino acid sequence aligned to the crystal structure. Similar studies for evaluating structures generated by ColabFold [Mirdita et al., 2022], a faster version of AlphaFold2 [Jumper et al., 2021], can be found in Appendix E. However, although the model has never seen ColabFold structures during training, the performance is similar to ESMFold structures. Further, we will also be evaluating our model on the CrossDocked 2020 dataset [Francoeur et al., 2020]. This dataset contains similar binding pockets, with different ligands docked to these pockets, and is sometimes used to train docking algorithms [McNutt et al., 2021].

**Metrics.** To evaluate the quality of a docking prediction, we can compare how much the predicted ligand pose differs from the ground truth position. Commonly, the root mean squared deviation (RMSD) of the predicted and ground truth ligand atom position pairs is used for that. A pose prediction with an *RMSD below* 2Å is considered to be approximately correct [Alhossary et al., 2015; Hassan et al., 2017; McNutt et al., 2021], so we calculate the percentage of predictions under this threshold. We also compare the *median RMSD* of the predictions for a better grasp of their quality. To evaluate the predictions of the sidechain atoms, we rely on a similar metric, namely the RMSD of the sidechain atoms (or SC-RMSD) to the ground truth holo crystal structure. Since we consider the backbone rigid, the sidechain atoms show less variation than the ligand and typically do not exceed a SC-RMSD of 4. Hence, we decided to use an SC-RMSD threshold of 1 for the main comparisons instead, but also show results for different thresholds (see Appendix F).

In all cases, even when using computationally generated structures as input, the holo crystal structure of the PDBBind dataset is always considered the ground truth. However, it is important to note that *in-silico* generated structures are often considerably different from the ground truth (compare Figure 10). A perfect match is thus unrealistic, especially for the SC-RMSD, as the conformations also differ in bond lengths. To compensate for this fact, we introduce a relative measure that compares the SC-RMSD before and after the prediction.

**Docking performance.** We compare our model to the freely available state-of-the-art search-based method GNINA and SMINA which outperform VINA [Koes et al., 2013a] on known binding-sites, and the diffusion-based model DIFFDOCK. Deep-learning methods with receptor flexibility [Qiao et al., 2023; McPartlon & Xu, 2023] are not available as of writing. Results are shown in Table 1. Our model is evaluated for drawing 10 and 40 samples, where we present metrics for the top-1 prediction, which corresponds to the highest-ranked prediction from the confidence model, as well as for the top-5 predictions, which selects the most accurate pose from the five highest-ranked predictions.

Our approach outperforms both search-based methods and DIFFDOCK in all instances, even when only drawing 10 samples. For bound protein docking with predicting 40 samples, we achieve an

approximately correct docking pose in $49.8\%$ of instances. In rigid docking, GNINA also performs well in this task, achieving $42.7\%$, but no other compared method with flexibility is competitive at this benchmark ($27.8\%$). We can see that current methods suffer from a substantial loss in docking accuracy when introducing flexibility while also requiring significantly more time to predict poses (and sidechains). We attribute this to the fact that the search space grows exponentially with each atom position, which limits search-based approaches.

Table 1: **PDBBind docking performance.** This table compares the performance of different docking methods on computationally generated structures and crystal structures. Methods that do not model the receptor as flexible, have been marked with the keyword rigid. All methods other than DIFFDOCK use site-specific docking and use the same pocket definition (i.e., the mean of C-$\alpha$ atoms within 5Å of any ligand atom). For a more detailed explanation of how these numbers were computed for existing approaches, see Appendix D. The numbers for the methods highlighted with a * were taken from Corso et al. [2023].

| | Apo ESMFold Proteins | | | | Holo Crystal Proteins | | | | |
| | Top-1 RMSD | | Top-5 RMSD | | Top-1 RMSD | | Top-5 RMSD | | Average |
| Method | %<2 | Med. | %<2 | Med. | %<2 | Med. | %<2 | Med. | Runtime (s) |
|---|---|---|---|---|---|---|---|---|---|
| DIFFDOCK (blind, rigid)* | 20.3 | 5.1 | 31.3 | 3.3 | 38.2 | 3.3 | 44.7 | 2.4 | 40 |
| SMINA (rigid) | 6.6 | 7.7 | 15.7 | 5.6 | 32.5 | 4.5 | 46.4 | 2.2 | 258 |
| SMINA | 3.6 | 7.3 | 13.0 | 4.8 | 19.8 | 5.4 | 34.0 | 3.1 | 1914 |
| GNINA (rigid) | 9.7 | 7.5 | 19.1 | 5.2 | 42.7 | 2.5 | 55.3 | 1.8 | 260 |
| GNINA | 6.6 | 7.2 | 12.1 | 5.0 | 27.8 | 4.6 | 41.7 | 2.7 | 1575 |
| DIFFDOCK-POCKET (10) | 41.0 | **2.6** | 47.6 | 2.2 | 47.7 | 2.1 | 56.3 | 1.8 | **17** |
| DIFFDOCK-POCKET (40) | **41.7** | **2.6** | **47.8** | **2.1** | **49.8** | **2.0** | **59.3** | **1.7** | 61 |

Furthermore, when docking to computationally generated structures, we achieve four times higher results as the best search-based method GNINA and nearly double the previous state-of-the-art DIFFDOCK on top-1 predictions. When run on GPU hardware, our model is also significantly faster than search-based methods (especially with flexibility modeling turned on). This can be extremely useful for practitioners because this allows them to use DIFFDOCK-POCKET for high-throughput tasks, even when the experimental structures are unavailable.

**Sidechain prediction quality.** All flexible methods investigated predict the sidechain positions jointly with the ligand pose. For predictions on the ground-truth, the sidechains are first randomly perturbed by the respective methods. We now investigate the quality of these predictions for SMINA and GNINA (we do not compare to DIFFDOCK as it is unable to model flexible residues). Table 2 illustrates the performance similarly to the docking results. Both SMINA and GNINA fail to predict accurate sidechains for computationally generated structures and crystal structures. Our approach achieves good sidechain reconstruction in $33.3\%$ and $49.2\%$ of cases for computationally generated structures and crystal structures respectively.

Table 2: **PDBBind sidechain performance.** Comparing the predicted sidechains of the different models with different inputs to the ground truth crystal structures.

| | Apo ESMFold Proteins | | | | Holo Crystal Proteins | | | |
| | Top-1 SC-RMSD | | Top-5 SC-RMSD | | Top-1 SC-RMSD | | Top-5 SC-RMSD | |
| Method | %<1 | Med. | %<1 | Med. | %<1 | Med. | %<1 | Med. |
|---|---|---|---|---|---|---|---|---|
| SMINA | 0.6 | 2.4 | 1.8 | 2.0 | 4.7 | 1.8 | 8.3 | 1.4 |
| GNINA | 0.6 | 2.5 | 1.8 | 2.0 | 3.3 | 1.7 | 7.7 | 1.4 |
| DIFFDOCK-POCKET (10) | **33.3** | **1.2** | **44.6** | **1.1** | **49.2** | **1.0** | 58.6 | **0.9** |
| DIFFDOCK-POCKET (40) | 32.6 | **1.2** | 44.4 | **1.1** | 48.7 | **1.0** | **59.2** | **0.9** |

The *in-silico* generated structures already have a median SC-RMSD of 1.5Å and $20.5\%$ of structures have an SC-RMSD of less than 1Å. This means that the sidechain predictions of SMINA and GNINA are worse than those of structure generation algorithms without access to information about the ligand. This becomes more apparent when investigating these numbers visually in Figure 4. Both score-based methods improve the sidechains only in less than $10\%$ of cases. Overall, DIFFDOCK-POCKET predicts sidechains that are substantially closer to the ground truth.

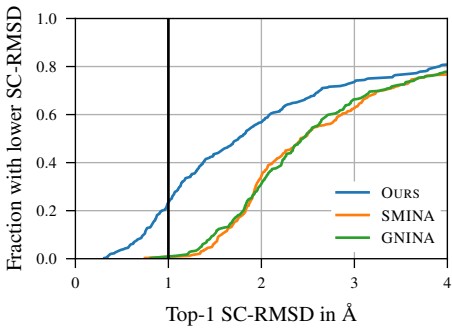 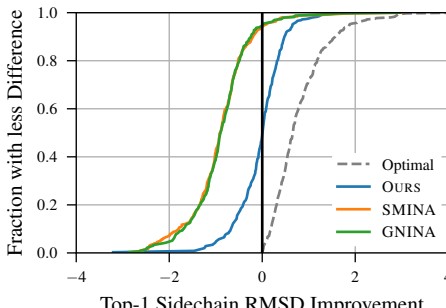

Figure 4: **Quality of predicted sidechains for *in-silico* structures.** *Left*: The cumulative distribution function shows how many instances have an SC-RMSD below a certain threshold to the holo structure. *Right*: The relative SC-RMSD between the structures before and after the predictions. The optimal line is computed by conformer matching the *in-silico* structures to the crystal structure.

**Cross-docking performance.** To demonstrate that the model can generalize to different scenarios, we evaluated it on the task of pocket-level cross-docking, as seen in Table 3. Our model achieves a pocket-normalized RMSD of less than 2Å in $28.6\%$ of instances, compared to the best other method of $24.4\%$. As for the overall accuracy, GNINA yields the best results. Brocidiacono et al. [2023] argue that the pocket-normalized score is more important since the size of the dockings per pocket is unevenly distributed. These results for our model are especially impressive considering that a) cross-docked structures were never seen during training, but some of the other approaches trained with this data, and b) the definition of the pocket center was out of distribution for our model. When we use the available data but compute the center of the pocket the same way as we did during training, our model achieves substantially higher results (compare Section F.7). This benchmark shows that DIFFDOCK-POCKET generalizes well to unseen structures and is suitable for a wide range of tasks.

Table 3: **Cross-docking performance on CrossDocked 2020.** Evaluation of the top-1 RMSD between different methods on the CrossDocked 2020 testset with complexes removed that were seen during training. The pocket-normalized percentage is presented for each value, and the overall score is listed in brackets. For the pocket-normalized score, the average performance on each pocket is reported instead of the performance across all complexes. Numbers for the methods marked with a * were taken from Brocidiacono et al. [2023].

| | Top-1 RMSD | | Average |
|---|---|---|---|
| Method | %<2 | %<5 | Runtime (s) |
| VINA* | 11.7 (15.6) | 40.2 (37.9) | 73.7 |
| GNINA* | 21.5 (**23.5**) | 51.7 (47.3) | 51.6 |
| DIFFDOCK* (blind) | 17.3 (11.6) | 51.7 (47.3) | 98.7 |
| PLANTAIN* | 24.4 (15.2) | **73.7 (71.9)** | **4.9** |
| DIFFDOCK-POCKET (10) | 28.3 (17.7) | 67.5 (50.2) | 22.0 |
| DIFFDOCK-POCKET (40) | **28.6** (18.5) | 67.9 (49.4) | 87.2 |

## 5 CONCLUSION

In this paper, we presented DIFFDOCK-POCKET, a fast diffusion-based generative model to dock small molecules. In contrast to many other ML-based approaches, we are able to incorporate prior knowledge of the binding pocket and model the protein's sidechain atoms close to the binding site as flexible. Our approach improves the state-of-the-art in almost all tested instances while also being significantly faster. Traditional approaches exhibit a drastic decline in runtime and accuracy when modeling receptor flexibility, which is not the case for our approach. A similar trend can be observed when using computationally generated structures, with which our approach works exceptionally well and loses almost no accuracy. Even when presenting the model with out-of-distribution data and pockets, our model improves the score for the pocket-normalized RMSD for CrossDocked2020 compared to existing methods. Especially in combination with *in-silico* generated structures, which can be generated quickly, we believe that our model opens new capabilities in high-throughput tasks, such as drug screening.

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

## A  BOUND ON REDUCED PREDICTION SPACE

As mentioned in the main text, our model makes predictions in a reduced, lower-dimensional space instead of predicting all atom positions. We can assess the reduction by counting the degrees of freedom of translations on the ligand and flexible sidechains as a function of their number of atoms. Sidechains have $m - r$ degrees of freedom for $m$ atoms on $r$ residues, since each residue has $m_r - 1$ torsion angles (where $m_r$ is the number of atoms in one residue). Since the maximum number of torsional angles in an amino acid (counted by our algorithm) is five, we can further bound $m - r$ with $0.8m$. Similarly, we can bound the ligand degrees of freedom by $n - 2 + 6$, 6 for the freedom of rotations and translations, and $n - 2$ the degrees of freedom from the torsion angles. This is because we can use an upper bound by assuming a tree-like bond structure between the ligand atoms, which means $n - 1$ bonds for $n$ atoms and, therefore $n - 2$ degrees of freedom (in case there is a cycle the ligand graph would have one more bond but it would also lose a degree of freedom from the restriction of the cycle structure). We can then compare the dimensions of $0.8m + n + 4$ to $3(m + n)$ and conclude that the three-dimensional coordinate space clearly has magnitudes larger (about three times as many) degrees of freedom, already for molecules with a small number of atoms.

## B  STERIC CLASHES

Steric clashes play a fundamental role in molecular interactions and structural biology. These clashes occur when atoms, or groups of atoms, come too close to each other, resulting in repulsive forces that hinder their ability to adopt certain conformations. In the context of generative modeling of complex structures, these clashes occur when atoms or groups of atoms in a three-dimensional structure are placed too closely together, violating the principles of molecular geometry and leading to unfavorable interactions. In essence, steric clashes represent a clash of physical space, as atoms cannot occupy the same space simultaneously due to their electron clouds. Understanding and mitigating steric clashes are important to check in generative modeling because they can lead to the generation of incorrect or physically unrealistic structures.

To quantify and evaluate steric clashes, several computational methods have been developed. One common approach involves computing the overlap between van der Waals radii of atoms. The van der Waals radii represent the approximate size of atoms and are typically defined as the distance at which the attractive van der Waals forces balance the repulsive forces between two atoms. To detect steric clashes, we assessed whether the van der Waals radii of atoms or groups of atoms in a molecular structure overlap by at least $0.4$ Angstroms (Å). If the overlap exceeds this threshold, it indicates a steric clash, suggesting that the molecular conformation is unfavorable due to repulsive forces. For the concrete values, we followed the tables from Mantina et al. [2009].

### B.1  REDUCING STERIC CLASHES IN PROTEIN SIDECHAIN ALIGNMENT

To train our flexible model, we align the sidechains of the unbound (apo) ESMFold protein with the bound (holo) crystal structure with conformer matching. Especially in cases where the predicted atomic structure differs from the actual true structure, simply reducing the RMSD between those two structures might lead to unrealistic proteins. For example, there could be a lot of steric clashes or the sidechain atoms completely turned away from the pocket. We introduced an additional penalty term when aligning the two protein structures to overcome these issues. The term that produced the most reasonable outputs (with regard to the number of steric clashes) was

$$\text{RMSD}(\text{Crystal Sc}, \tilde{\text{Sc}}) \cdot \frac{\sqrt{\sum_{l \in Lig, s \in Sc} e^{-(s-l)^2}}}{\sqrt{\sum_{l \in Lig, s \in Sc} e^{-(s-l)^2}(s-l)^2}}. \tag{3}$$

$s$ and $l$ are the positions of atoms of the sidechains and ligands respectively.

We calculate the pairwise distances between the ligand and sidechain atoms, with an exponential weighting scheme applied to emphasize closer atoms of the protein. The weights are calculated based on the exponential of the negative distances, indicating a stronger penalty for closer atomic interactions. The resulting weighted distances are then summed and normalized, contributing to an overall penalty term incorporated into the calculation of the root-mean-square deviation (RMSD) of

the modified atoms. This RMSD, adjusted by the weighted penalty term, measures the structural deviation while accounting for steric clashes. The method reduces clashes by penalizing close atomic contacts and promoting greater separation between the ligand and protein, as seen in Table 4. While conformer matching already reduces the number of steric clashes, this penalty can further reduce the number. All RMSDs that are shown in this paper are calculated by removing the hydrogens and computing the distance between all atoms, not just the C-$\alpha$ backbone.

Table 4: **Steric clashes for *in-silico* structures.** This table analyzes the number of steric clashes between the receptor and the ligand.

| Method | Percentage with Steric Clashes | Average Number of Steric Clashes |
|---|---|---|
| Crystal structures | 14.3 | 0.2 |
| ESMFold2 structures | 76.7 | 19.1 |
| Conformer-Matched | 68.3 | 15.4 |
| Conformer-Matched w/ penalty | 67.7 | 13.9 |

## B.2 MODEL RESULTS

Given this definition of steric clashes, we can evaluate the different models, as done in Table 5. It can be seen that flexible models produce substantially more steric clashes, especially when executed on computationally generated structures. This aligns well with the fact that the ESMFold structure itself already exhibits many steric clashes. Our model produces more steric clashes than search-based methods on *in-silico* structures and drastically more on the crystal structure. For the ESMFold predictions, this may be because our model achieves more than four times the docking performance on this data, and the other methods typically predict wrong ligand poses, which are possibly far away (see high median RMSD). For example, SMINA predicts the least number of steric clashes, but also has the lowest docking performance. However, this table definitely highlights a shortcoming of our approach for at least crystal structures. Those shortcomings of ML docking methods have been discussed by Buttenschoen et al. [2023] and can be reduced by performing small optimizations of the predicted docking poses.

Table 5: **Steric clashes for top-1 predictions.** Comparison of the number of steric clashes between the receptor and ligand atoms using the predictions of different models and structures.

| Method | Apo ESMFold Proteins Percentage with Steric Clashes | Average Number of Steric Clashes | Holo Crystal Proteins Percentage with Steric Clashes | Average Number of Steric Clashes |
|---|---|---|---|---|
| SMINA (rigid) | 0.9 | 0.1 | 0.0 | 0.0 |
| SMINA | 60.4 | 12.8 | 1.1 | 0.0 |
| GNINA (rigid) | 5.4 | 0.4 | 1.7 | 0.1 |
| GNINA | 52.7 | 12.7 | 0.3 | 0.0 |
| DIFFDOCK-POCKET (10) | 69.3 | 9.8 | 57.7 | 4.4 |
| DIFFDOCK-POCKET (40) | 69.0 | 9.2 | 55.3 | 4.1 |

## C MODEL DETAILS

### C.1 ARCHITECTURE

The protein and the ligand structures can be represented as geometric graphs. Our architecture uses three different graphs: a graph containing the ligand atoms, one that contains the protein atoms, and a third where each node corresponds to a residue (i.e., an amino acid). The atom nodes of the ligands and proteins are featurized with their chemical properties, the residue nodes with embeddings of the ESM2 language model [Lin et al., 2023].

The nodes in each graph are connected to nodes in the same graph with inter-graph edges. We construct receptor-receptor and residue-residue edges between an atom and its $k$ nearest neighbors (for residues we use the C-$\alpha$ positions). The ligand-ligand edges correspond to bonds between the

ligand atoms that are featurized by their bonding type, and additionally, we form edges between atoms under a cutoff distance of 5Å.

Nodes can also be connected to nodes in the other graphs by (dynamic) cross edges. For the ligand-receptor and ligand-atom edges, we form edges between atoms based on a distance threshold that is calculated with the diffusion noise. As for the atom-residue graphs, we connect each residue to the atoms it consists of. As the positions of the ligand and receptor atoms are dynamic in the diffusion process, these graphs need to be reconstructed at each time step.

Several convolutional layers are concatenated in which the nodes pass messages using tensor products based on the node features and irreducible representations of the edges. The number of convolutional layers differs between the score and confidence model. MLPs are then used on the node embeddings to make the final predictions.

## C.2   Training the Confidence Model

To train the confidence model, we trained a smaller score model (in the same way as the main/large model) that predicts more diverse but less accurate ligand poses and protein structures. The predictions are then evaluated against the ground truth to create a label that indicates whether the RMSD is $< 2$Å and the RMSD of the flexible atoms in the sidechains is $< 1$Å. The confidence model then learns to predict a label of $1$ iff the prediction of the score model is good in terms of docking and sidechain atom positions. The model is then trained with a binary cross-entropy loss. No diffusion is involved in the training of the confidence model.

## C.3   Sidechain Flexibility

The flexible residues can be automatically determined based on the distance to the ground truth ligand pose or, at inference, manually specified when there is no access to a ground truth ligand. We then select residues with atoms inside a rectangular prism around the ligand as also used in previous works [McNutt et al., 2021]. This means that with a "radius" of $r$ every residue is selected where for the coordinates $x, y, z$ any atom of this amino acids it holds that

$$
\begin{aligned}
\min(lig_x) - r &< x < \max(lig_x) + r \\
\min(lig_y) - r &< y < \max(lig_y) + r \\
\min(lig_z) - r &< z < \max(lig_z) + r,
\end{aligned}
$$
(4)

where $lig_x$, $lig_y$ and $lig_z$ mean the collection of $x$, $y$ and $z$ coordinates of the ligand atoms. This defines a prism around the ligand with an additional radius $r$. For a flexible radius, we chose 3.5Å as modeling flexibility for sidechains within this radius to the ligand was found to be a reasonable representation for structural changes happening upon ligand binding in Meli et al. [2021]. During inference, we cannot assume to have any information regarding the ligand position therefore instead of calculating a prism around the ligand, the user needs to set them manually.

To determine the concrete bonds at which torsional angles need to be applied, we build a graph for each amino acid according to the chemical structure. Each found rotatable bond is stored as the corresponding edge and subgraph that starts at the second vertex/end of the edge, onto which a rotation would be applied. See Algorithm 1 for the implementation.

Corso et al. [2023] had to rely on the definition of preservation of angular velocity and Kabsch alignments to disentangle the effect of the updates in torsion angles of the ligands from the roto-translation of the ligand w.r.t. the protein. In our case, we keep this convention for the disentanglement of the degrees of freedom of the ligand. When it comes to defining the direction of update of the torsion angles of the sidechains of the protein, we always rotate the side that does not contain the protein backbone. This simple convention makes the update of the sidechain's conformation disentangled from the roto-translation of the ligand w.r.t. the protein without requiring any additional Kabsch alignment. We note that in practice this is very similar to the induction of no linear or angular velocity in the protein due to the significantly larger size of the rest of the protein compared to the individual sidechain.

---

**Algorithm 1:** Graph Traversal to Compute Rotatable Bonds

---

**Input:** Atom positions $x$, atom names $\mathcal{N}$
**Output:** Rotable bonds $\mathcal{B}$, rotation mask $\mathcal{M}$
$(x, \mathcal{N}) \leftarrow$ `removeHydrogens`$(x, \mathcal{N})$;
$G \leftarrow$ `constructDirectedGraph`$(x, \mathcal{N})$;
**for** $e \in edges($`BFS`$(G))$ **do**
    $G_U \leftarrow$ `toUndirected`$(G)$;
    $G_U \leftarrow$ `removeEdge`$(G_U, e)$;
    **if** *not* `isConnected`$(G_U)$ **then**
        $c \leftarrow$ `connectedComponents`$(G_U)$;
        **if** `size`$(sorted(c)[0]) > 1$ **then**
            $\mathcal{M}.append(c[1])$;
            $\mathcal{B}.append(e)$;
        **end**
    **end**
**end**

---

## C.4 Training and Inference of the Score Model

We use ESMFold2 predicted structures conformer-matched to the PDBBind crystal structures to train the score model. If the RMSD in the pocket between the ground truth and *in-silico* structure is larger than 2Å, we assume that ESMFold was unable to predict a good structure and use the ground truth holo structure instead. The training and inference procedures were inspired by DIFFDOCK and can be seen in Algorithm 2 and Algorithm 3 respectively.

At inference (i.e., Algorithm 3), it is important to note that the model is not aware of any of the ground-truth ligand or sidechain positions. As such, there is no possibility for data leakage as the model is neither aware of the ground-truth sidechain positions, nor which sidechains are flexible.

---

**Algorithm 2:** Training Epoch

---

**Input:** Training pairs: $\{(\mathbf{x}^\star, \mathbf{y}^\star), \}$, flexibility radius: r, pocket radius: p with buffer
**foreach** $\mathbf{x}^\star, \mathbf{y}^\star$ **do**
    Let $\mathbf{x}_0 \leftarrow \arg\min_{\mathbf{x}^\dagger \in \mathcal{M}_{tr,rot,tor,\mathbf{x}^\star}} \text{RMSD}(\mathbf{x}^\star, \mathbf{x}^\dagger)$;
    Let
    $\tilde{\mathbf{y}}^\star \leftarrow \{res \in \mathbf{y}^\star : \exists \, atom = (a_x, a_y, a_z) \in res, a_x \in [\min_x(\mathbf{x}^\star) - \mathbf{r}, \max_x(\mathbf{x}^\star) + \mathbf{r}], a_y \in$
    $[\min_y(\mathbf{x}^\star) - \mathbf{r}, \max_y(\mathbf{x}^\star) + \mathbf{r}], a_z \in [\min_z(\mathbf{x}^\star) - \mathbf{r}, \max_z(\mathbf{x}^\star) + \mathbf{r}]\}$;
    Let $\mathbf{y}_0^\star \leftarrow \arg\min_{\mathbf{y}^\dagger \in \mathcal{M}_{sc-tor,\mathbf{y}^\star}} \text{RMSD}(\tilde{\mathbf{y}}^\star, \tilde{\mathbf{y}}^\dagger) \cdot \text{penalty}$;
    Let pocket center = $pc \leftarrow$ average of positions of $C_\alpha \in \{$residue $\in \mathbf{y}^\star \, \exists atom = a \in$
    residue for which $\exists$ ligand atom $l \in \mathbf{x}_0 \| a - l \| < p\}$ if the set is empty, then closest $C_\alpha$;
    Let $\mathbf{y}_0 \leftarrow \{$res $\in \mathbf{y}_0^\star : \exists a \in$ res for which $\exists l \in \mathbf{x}_0 : \| a - l \| < \text{circumradius}(\mathbf{y}_0^\star) + \text{buffer}\}$;
    Sample $t \sim \mathcal{U}([0, 1])$;
    Sample $\Delta\mathbf{r}, \Delta R, \Delta\boldsymbol{\theta}_l, \Delta\boldsymbol{\theta}_{sc}$, from diffusion kernels
    $p_t^{\text{tr}}(\cdot \mid 0), p_t^{\text{rot}}(\cdot \mid 0), p_t^{\text{tor}_l}(\cdot \mid 0), p_t^{\text{tor}_{sc}}(\cdot \mid 0)$;
    Compute $\mathbf{x}_t$ by applying $\Delta\mathbf{r}, \Delta R, \Delta\boldsymbol{\theta_l}$ to $\mathbf{x}_0$;
    Compute $\mathbf{y}_t$ by applying $\boldsymbol{\theta_{sc}}$ to $\tilde{\mathbf{y}}_0$;
    Predict scores $\alpha \in \mathbb{R}^3, \beta \in \mathbb{R}^3, \gamma \in \mathbb{R}^n, \delta \in \mathbb{R}^m = \mathbf{s}(\mathbf{x}_t, \mathbf{y}_t, t)$;
    Take optimization step on loss
    $\mathcal{L} = ||\alpha - \nabla \log p_t^{\text{tr}}(\Delta\mathbf{r} \mid 0)||^2 + ||\beta - \nabla \log p_t^{\text{rot}}(\Delta R \mid 0)||^2 +$
    $||\gamma - \nabla \log p_t^{\text{tor}_l}(\Delta\boldsymbol{\theta_l} \mid 0)||^2 + ||\delta - \nabla \log p_t^{\text{tor}_{sc}}(\Delta\boldsymbol{\theta_{sc}} \mid 0)||^2$
**end**

---

---

**Algorithm 3:** Inference Algorithm

---

**Input:** RDKit prediction $\mathbf{c}$, generated protein structure $\mathbf{d}$, flxibility radius r, pocket radius p
  with buffer (both centered at origin)
**Output:** Sampled ligand pose $\mathbf{x}_0$, sampled protein pose $\mathbf{y}_0$ with applied pocket knowledge
Let pocket center = $pc \leftarrow$ average of positions of $C_\alpha \in \{$residue $\in \mathbf{d}$ $\exists$atom $= a \in$
  residue for which $\exists$ ligand atom $l \in \mathbf{c}\|a - l\| < p\}$;
Let $\mathbf{d}^\star \leftarrow \{res \in \mathbf{d} : \exists a \in res, \|a - pc\| < $ circumradius$(\mathbf{c}) + $ buffer$\}$;
Sample $\boldsymbol{\theta}_{l;N} \sim \mathcal{U}(SO(2)^k)$, $R_N \sim \mathcal{U}(SO(3))$, $\mathbf{r}_N \sim \mathcal{N}(0, \sigma_{\text{tr}}^2(T))$ $\boldsymbol{\theta}_{sc,N} \sim \mathcal{U}(SO(2)^m)$;
Define $\tilde{\mathbf{y}}_k$ from $\mathbf{y}_k$ as $\{$residue $= res \in \mathbf{y}_k : \exists$atom $= a \in res, \|a - pc\| < $ r$\}$;
Randomize ligand and sidechains by applying $\mathbf{r}_N, R_N, \boldsymbol{\theta}_{l;N}$, to $\mathbf{c}$ and $\boldsymbol{\theta}_{sc;N}$ to $\tilde{\mathbf{d}}^\star$;
**for** $n \leftarrow N$ **to** $1$ **do**
  Let $t = n/N$ and $\Delta\sigma_{\text{tr}}^2 = \sigma_{\text{tr}}^2(n/N) - \sigma_{\text{tr}}^2((n-1)/N)$ and similarly for
    $\Delta\sigma_{\text{rot}}^2, \Delta\sigma_{\text{tor}_l}^2, \Delta\sigma_{\text{tor}_{sc}}^2$;
  Predict scores $\alpha \in \mathbb{R}^3, \beta \in \mathbb{R}^3, \gamma \in \mathbb{R}^k, \delta \in \mathbb{R}^m, \leftarrow \mathbf{s}(\mathbf{x}_n, \mathbf{y}_n, t)$;
  Sample $\mathbf{z}_{\text{tr}}, \mathbf{z}_{\text{rot}}, \mathbf{z}_{\text{tor}_l}, \mathbf{z}_{\text{tor}_{sc}}$ from $\mathcal{N}(0, \Delta\sigma_{\text{tr}}^2), \mathcal{N}(0, \Delta\sigma_{\text{rot}}^2), \mathcal{N}(0, \Delta\sigma_{\text{tor}_l}^2), \mathcal{N}(0, \Delta\sigma_{\text{tor}_{sc}}^2)$
    respectively;
  Set $\Delta\mathbf{r} \leftarrow \Delta\sigma_{\text{tr}}^2\alpha + \mathbf{z}_{\text{tr}}$ and similarly for $\Delta R, \Delta\boldsymbol{\theta}_l, \Delta\boldsymbol{\theta}_{sc}$;
  Compute $\mathbf{x}_{n-1}$ by applying $\Delta\mathbf{r}, \Delta R, \Delta\boldsymbol{\theta}_l$, to $\mathbf{x}_n$;
  Compute $\mathbf{y}_{n-1}$ by applying $\Delta\boldsymbol{\theta}_{sc}$, to $\tilde{\mathbf{y}}_n$;
**end**
Return $\mathbf{x}_0, \mathbf{y}_0$;

---

# D BENCHMARKING DETAILS

In our experimentation, we used NVIDIA RTX 6000 GPUs to conduct the assessment of our model's performance. To ensure robustness and reliability, we executed the model three times, each run initiated with seeds 0, 1, and 2. It is crucial to note that while seeds were employed to initialize the runs, achieving 100 percent reproducibility proved challenging due to the inherent non-deterministic nature of certain operations when executed on a GPU. To enhance the reliability of our reported values, we computed the mean across the three runs, providing a more stable and indicative measure of the model's performance rather than relying on individual figures from a single run. This approach ensures that our reported results reflect the averaged behavior of the model under different seed initializations, acknowledging and addressing the inherent stochasticity introduced by GPU computations.

## D.1 PARAMETERS FOR GNINA AND SMINA

We opted to use the default/suggested parameters as much as possible when running GNINA and SMINA. We set the exhaustiveness (number of Monte Carlo chains for searching) to 8. When applying the flexible features we chose the flexible radius to be 3.5Å as in our model, where GNINA also specifies the flexible sidechains as we do during training with a rectangular prism. We generated 10 modes for each run on which we were able to evaluate top-N metrics and provide a fair assessment accounting for the variance of the results of the algorithm.

For site-specific docking, GNINA has two distinct approaches. The first method involves establishing a rectangular prism around the ground truth atom, utilizing the minimum and maximum values for the x, y, and z coordinates. This prism can be further customized with the addition of a buffer (and in case the box defined by the prism is too small, it is appended in such a way that the ligand can rotate inside of it). Alternatively, the second method permits the construction of a Cartesian box by directly specifying the coordinates. In our comparative analysis with our results, we opted for the Cartesian box approach, as it aligns more closely with our definition of the ligand-binding pocket. This choice was also motivated by the perception that the prism method, relying on knowledge of the original ligand position, may introduce strong bias. However, even when using the autobox method to level the playing field, our results demonstrate that the performance of our model remains competitive. In this case, we compared the different approaches using the rigid model on crystal structures of the testset of PDBBind depicted in Table 6.

Even with no additional buffer when autoboxing the ligand, we can see that the results of GNINA are below 50% on the pre-processed files. We can also see that even doubling the exhaustiveness does not significantly affect the docking results. This plateau effect may indicate that the algorithm has adequately explored the conformational space, and additional computational resources do not lead to a proportional enhancement in the quality of predictions. When looking at the results of the preprocessed and original protein files, we can also observe that minor changes in the protein structure inputs result in significant differences in docking performance, suggesting a concerning sensitivity to variations in molecular configurations. This sensitivity is undesirable, especially when handling generated protein structures is a goal.

Clearly, the case of only autoboxing the ligand with no additional buffer does not reflect reality as the user would have to know the exact bounding box of the ligand with a 0Å margin of error. We can then observe that with an increase in the search space, the docking performance of GNINA deteriorates. The Cartesian pocket we selected exhibits very similar performance to the default setting, which incorporates a 4Å buffer through autoboxing, with only a marginal 1-2% difference. This justifies our comparison to the Cartesian box instead of the default GNINA settings while also being fair in having a similar pocket definition.

Table 6: **GNINA results with different attributes.** In this table, we present additional results for benchmarking GNINA: the differences in results with differently defined or sized pockets, exhaustiveness and input protein files.

| | | preprocessed PDB files | | | | on original PDB files | | | |
| | | Top-1 RMSD | | Top-5 RMSD | | Top-1 RMSD | | Top-5 RMSD | |
| Pocket Type | Exhaustiveness | <2% | Median | <2% | Median | <2% | Median | <2% | Median |
|---|---|---|---|---|---|---|---|---|---|
| Our pocket center + 10Å | 8 | 42.7 | 2.5 | 55.3 | 1.8 | 48.2 | 2.2 | 63.0 | 1.5 |
| Autobox Ligand + 0Å | 8 | **48.0** | **2.2** | 63.9 | **1.5** | **53.0** | **1.9** | **69.8** | **1.3** |
| | 16 | 45.7 | **2.2** | **85.6** | **1.5** | - | - | - | - |
| Autobox Ligand + 4Å | 8 | 43.6 | 2.3 | 58.1 | 1.7 | 51.0 | **1.9** | 67.2 | **1.3** |
| | 16 | 46.4 | **2.2** | 60.4 | 1.6 | - | - | - | - |
| Autobox Ligand + 10Å | 8 | 39.6 | 3.0 | 49.9 | 2.0 | 47.0 | 2.3 | 61.5 | 1.5 |
| | 16 | 42.2 | 2.7 | 54.7 | 1.8 | - | - | - | - |

## E    PERFORMANCE ON COLABFOLD

ColabFold [Mirdita et al., 2022] is a faster version of AlphaFold2 [Jumper et al., 2021] and is often used to generate a 3D structure based on a given sequence. In this part, we show how the model behaves on these structures instead of using ESMFold2 structures. This study is crucial since the model uses ESMFold embeddings during training for all proteins, and some of the training set also consists of high-quality structures predicted by ESMFold. This could mean that the model only works well with those specific structures while producing inferior results otherwise. To answer this, we have presented similar studies for ColabFold structures in Table 7, Table 8, and Table 9. We can see that the results are similar to those from ESMFold, letting us conclude that the model generalizes to well.

Table 7: **PDBBind docking performance with ColabFold structures.** Comparing the top-1 and top-5 results of multiple docking approaches when using structures generated by ColabFold.

| | Apo ColabFold Proteins | | | |
| | Top-1 RMSD | | Top-5 RMSD | |
| Method | %<2 | Med. | %<2 | Med. |
|---|---|---|---|---|
| SMINA (rigid) | 5.7 | 7.5 | 13.1 | 5.5 |
| SMINA | 5.3 | 7.0 | 11.5 | 5.4 |
| GNINA (rigid) | 10.5 | 7.3 | 18.0 | 5.0 |
| GNINA | 7.7 | 6.8 | 15.6 | 4.9 |
| DIFFDOCK-POCKET (10) | 37.5 | 2.8 | 45.0 | 2.3 |
| DIFFDOCK-POCKET (40) | **39.5** | **2.7** | **46.0** | **2.2** |

Table 8: **Top-1 PDBBind docking with ColabFold structures.** More detailed performance evaluation when docking to *in-silico* structures generated by ColabFold.

| | Ligand RMSD | | | | | Sidechain RMSD | | | | |
| | Percentiles ↓ | | | % below threshold ↑ | | Percentiles ↓ | | | % below threshold ↑ | |
| Methods | 25th | 50th | 75th | 2 Å | 5 Å | 25th | 50th | 75th | 1 Å | 2 Å |
|---|---|---|---|---|---|---|---|---|---|---|
| SMINA (rigid) | 5.1 | 7.5 | 11.4 | 5.7 | 23.9 | - | - | - | - | - |
| SMINA | 5.0 | 7.0 | 9.7 | 5.3 | 25.6 | 1.9 | 2.3 | 3.2 | 0.6 | 32.1 |
| GNINA (rigid) | 3.7 | 7.3 | 11.6 | 10.5 | 34.8 | - | - | - | - | - |
| GNINA | 4.1 | 6.8 | 10.3 | 7.7 | 33.5 | 1.9 | 2.3 | 3.1 | 0.3 | 32.9 |
| DIFFDOCK-POCKET (10) | **1.5** | 2.8 | **5.0** | 37.5 | **75.2** | **1.0** | **1.4** | **1.9** | **28.2** | **79.0** |
| DIFFDOCK-POCKET (40) | **1.5** | **2.7** | **5.0** | **39.5** | 74.6 | **1.0** | **1.4** | **1.9** | 27.6 | **79.0** |

Table 9: **PDBBind sidechain performance with ColabFold structures.** Evaluating the performance of the sidechains when relying on *in-silico* structures generated by ColabFold.

| | Apo ColabFold Proteins | | | |
| | Top-1 SC-RMSD | | Top-5 SC-RMSD | |
| Method | %<1 | Med. | %<1 | Med. |
|---|---|---|---|---|
| SMINA | 0.6 | 2.3 | 0.6 | 2.0 |
| GNINA | 0.3 | 2.3 | 1.2 | 1.9 |
| DIFFDOCK-POCKET (10) | **28.2** | **1.4** | **35.1** | **1.2** |
| DIFFDOCK-POCKET (40) | 27.6 | **1.4** | 34.9 | **1.2** |

# F ADDITIONAL RESULTS

## F.1 FURTHER DOCKING RESULTS

We have compiled a list of tables and figures that allow further evaluation of the docking results. In Table 10 and Table 11, we illustrate the different percentiles of our predictions for the ligand and sidechain predictions for both crystal structures and ESMFold. We also evaluate the models on a subset of the testset where UnitProt IDs that are present in the training or validation set have been removed. The results are shown in Table 12. Figure 5 shows the cumulative distribution functions of the top-1 docking RMSD.

Similarly as for the ligand docking accuracy, we also provide further studies for the sidechain accuracy. Figure 6 illustrates the fraction of predictions with a lower sidechain RMSD for crystal structures and ESMFold structures respectively. Since the sidechains of ESMFold structures cannot be aligned completely to the crystal structures by only changing the torsional angles, Figure 7 shows further studies on the relative SC-RMSD. The relative SC-RMSD is computed by subtracting the SC-RMSD of the ESMFold structure from the SC-RMSD of the predicted protein.

Table 10: **Top-1 PDBBind crystal docking.** A more detailed performance evaluation of docking with holo crystal structures.

| | Ligand RMSD | | | | | Sidechain RMSD | | | | |
| | Percentiles ↓ | | | % below Threshold ↑ | | Percentiles ↓ | | | % below Threshold ↑ | |
| Methods | 25th | 50th | 75th | 2 Å | 5 Å | 25th | 50th | 75th | 1 Å | 2 Å |
|---|---|---|---|---|---|---|---|---|---|---|
| SMINA (rigid) | 1.6 | 4.5 | 8.0 | 32.5 | 54.7 | - | - | - | - | - |
| SMINA | 2.8 | 5.4 | 7.8 | 19.8 | 47.9 | 1.6 | 1.8 | 2.2 | 2.0 | 63.8 |
| GNINA (rigid) | 1.2 | 2.5 | 6.8 | 42.7 | 67.0 | - | - | - | - | - |
| GNINA | 1.8 | 4.6 | 7.9 | 27.8 | 54.4 | 1.4 | 1.7 | 2.1 | 3.3 | 71.9 |
| DIFFDOCK-POCKET (10) | **1.1** | 2.1 | 4.5 | 47.7 | 78.7 | **0.6** | **1.0** | 1.6 | **49.2** | 85.7 |
| DIFFDOCK-POCKET (40) | **1.1** | **2.0** | **4.3** | **49.8** | **79.8** | **0.6** | **1.0** | **1.5** | 48.7 | **87.0** |

Table 11: **Top-1 PDBBind ESMFold docking.** A more detailed performance evaluation of docking with computationally generated ESMFold structures.

| | Ligand RMSD | | | | | Sidechain RMSD | | | | |
| | Percentiles ↓ | | | % below threshold ↑ | | Percentiles ↓ | | | % below threshold ↑ | |
| **Methods** | 25th | 50th | 75th | 2 Å | 5 Å | 25th | 50th | 75th | 1 Å | 2 Å |
|---|---|---|---|---|---|---|---|---|---|---|
| SMINA (rigid) | 5.4 | 7.7 | 11.9 | 6.6 | 22.5 | - | - | - | - | - |
| SMINA | 5.5 | 7.3 | 9.9 | 3.6 | 20.5 | 1.9 | 2.4 | 3.7 | 0.6 | 34.4 |
| GNINA (rigid) | 4.1 | 7.5 | 12.0 | 9.7 | 33.6 | - | - | - | - | - |
| GNINA | 4.7 | 7.2 | 10.5 | 6.6 | 28.0 | 1.9 | 2.5 | 3.7 | 0.6 | 31.0 |
| DIFFDOCK-POCKET (10) | 1.3 | **2.6** | 5.1 | 41.0 | 74.6 | **0.9** | **1.2** | **1.8** | **33.3** | 79.6 |
| DIFFDOCK-POCKET (40) | **1.2** | 2.6 | **5.0** | **41.7** | **74.9** | **0.9** | **1.2** | **1.8** | 32.6 | **80.3** |

Table 12: **Filtered PDBBind docking performance.** This table mirrors the resutls from Table 1, but has filtered out all the complexes of the testset where the UniProt ID appears in the training or validation set.

| | Apo ESMFold Proteins | | | | Holo Crystal Proteins | | | | |
| | Top-1 RMSD | | Top-5 RMSD | | Top-1 RMSD | | Top-5 RMSD | | Average |
| Method | %<2 | Med. | %<2 | Med. | %<2 | Med. | %<2 | Med. | Runtime (s) |
|---|---|---|---|---|---|---|---|---|---|
| DIFFDOCK (blind, rigid)* | - | - | - | - | 20.8 | 6.2 | 28.7 | 3.9 | 40 |
| SMINA (rigid) | 6.5 | 7.7 | 15.9 | 6.2 | 29.0 | 5.1 | 45.7 | 2.2 | 258 |
| SMINA | 4.8 | 7.6 | 12.7 | 5.3 | 18.3 | 6.2 | 38.7 | 3.0 | 1914 |
| GNINA (rigid) | 10.1 | 7.2 | 20.3 | 5.3 | **39.9** | 2.6 | **54.5** | **1.9** | 260 |
| GNINA | 8.7 | 6.6 | 15.9 | 4.9 | 24.8 | 4.5 | 38.7 | 2.9 | 1575 |
| DIFFDOCK-POCKET (10) | **27.7** | **3.3** | **34.6** | 2.8 | 36.5 | 2.5 | 49.4 | 2.0 | **17** |
| DIFFDOCK-POCKET (40) | 26.3 | **3.3** | 33.6 | **2.7** | 39.2 | **2.4** | 52.4 | **1.9** | 61 |

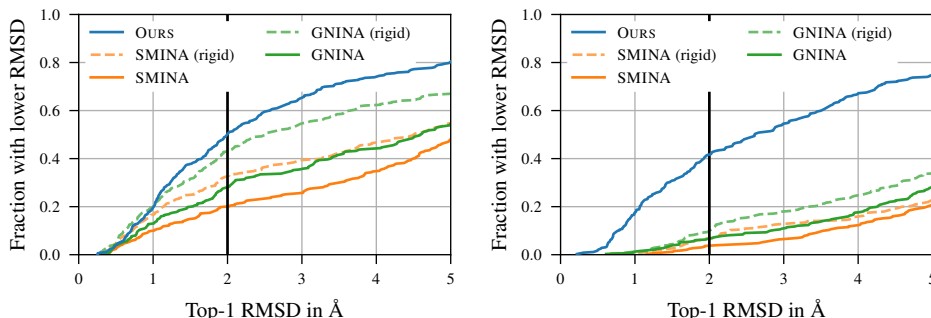

Figure 5: **Cumulative distribution function of RMSD.** *Left*: The CDF when using crystal structures as input. *Right*: The CDF when using ESMFold structures as input.

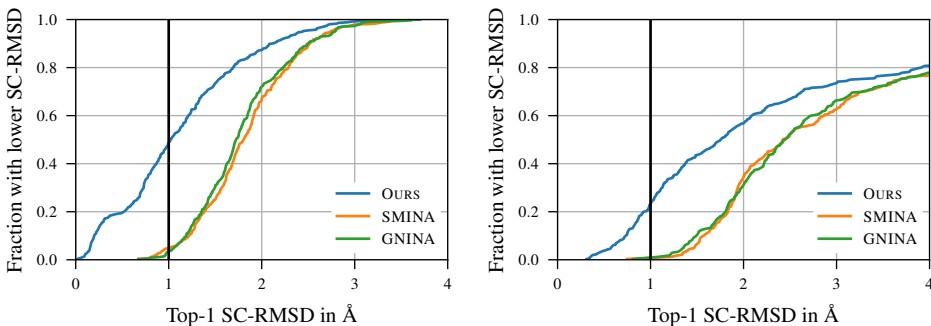

Figure 6: **Cumulative distribution function of SC-RMSD.** *Left*: The CDF when using crystal structures as input. *Right*: The CDF when using ESMFold structures as input.

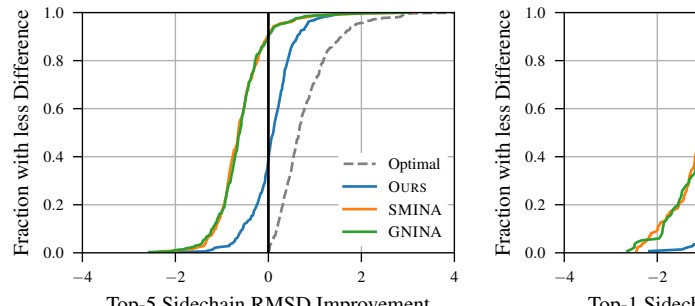

Figure 7: **Relative sidechain improvements on ESMFold structures.** *Left*: The relative sidechain improvement, when picking the top-5 sidechain prediction. *Right*: The relative sidechain improvement only for ESMFold complexes that have a pocket RMSD of $< 1.5$Å.

## F.2  RIGID MODEL COMPARISON

In this section, we will investigate the impact of training with flexibility on the model's performance. For this, we trained a rigid model on the holo crystal structure of proteins with pocket reduction, and compared it to a flexible model. In all cases, we used models without low-temperature sampling, 20 inference steps and 10 samples per complex. In Table 13 this comparison is illustrated. We further added a comparison for when we use the flexible model, but do not predict the pose of any sidechain positions during training.

From this, we can see that training with flexibility improves the docking accuracy, especially for proteins where the true sidechain conformations are unknown (i.e., apo). We can also see that the performance decreases when using a flexible model in a rigid fashion. However, in our experiments, these effects were less prominent when relying on low-temperature sampling.

Table 13: **PDBBind docking performance rigid and flexible.** We compare the docking performance of a rigid model, a model that was trained with flexibility (marked with *), and the same model but without flexibility at inference†. None of the models use low-temperature sampling.

|  | Apo ESMFold Proteins | | | | Holo Crystal Proteins | | | |
|  | Top-1 RMSD | | Top-5 RMSD | | Top-1 RMSD | | Top-5 RMSD | |
| Method | %<2 | Med. | %<2 | %<5 | %<2 | Med. | %<2 | %<5 |
|---|---|---|---|---|---|---|---|---|
| DIFFDOCK-POCKET (rigid) | 29.8 | 3.6 | 40.7 | 76.7 | 44.7 | 2.4 | 55.0 | 86.5 |
| DIFFDOCK-POCKET* | **37.7** | **3.0** | **45.9** | **82.2** | **45.4** | **2.2** | **57.2** | **87.6** |
| DIFFDOCK-POCKET† | 24.9 | 4.0 | 41.0 | 76.8 | 27.7 | 3.5 | 45.9 | 81.5 |

## F.3  PERFORMANCE ON MEMBRANE PROTEINS

Membrane proteins make up more than 60% of the drug targets in humans and hence play a crucial role in drug discovery [Overington et al., 2006]. In the testset of PDBBind, there are nine proteins that are membrane proteins that have been classified as such by either White [2009]; Lomize et al. [2011]; Kozma et al. [2012]; Newport et al. [2018]. The corresponding PDB ids are: 6e4v, 6h7d, 6iql, 6kqi, 6n4b, 6qxa, 6qzh, 6r7d, 6rz6. The docking performance of our model on these nine proteins is illustrated in Table 14. We can see that for experimentally generated crystal structure and ColabFold membrane proteins our model archives only in 33.3% of cases a ligand RMSD of $< 2$. For ESMFold, there is no successful docking for these proteins. We believe this is the case because the quality of the structure of ESMFold is worse on these proteins as ColabFold (compare Table 15).

Table 14: **Docking performance on PDBBind membrane proteins.** This table denotes the Top-1 ligand RMSD on the listed proteins for different protein structures.

| Protein Structure | Top-1 Ligand RMSD in Å | | | | | | | | |
|---|---|---|---|---|---|---|---|---|---|
| | 6e4v | 6h7d | 6iql | 6kqi | 6n4b | 6qxa | 6qzh | 6r7d | 6rz6 |
| Crystal | 8.8 | 1.6 | **2.2** | 3.5 | **1.2** | 13.7 | **2.3** | **4.8** | 2.0 |
| ESMFold | **5.6** | 3.4 | 3.1 | **2.3** | 5.3 | 10.5 | 5.1 | 9.0 | 2.7 |
| ColabFold | 6.3 | **1.5** | 2.9 | 3.6 | 1.7 | **10.0** | 5.5 | 5.4 | **1.9** |

Table 15: **Pocket RMSDs of PDBBind membrane proteins.** The RMSDs between the atoms of the receptor and the computationally generated protein are shown in this table.

| Protein Structure | Pocket RMSD in Å | | | | | | | | |
|---|---|---|---|---|---|---|---|---|---|
| | 6e4v | 6h7d | 6iql | 6kqi | 6n4b | 6qxa | 6qzh | 6r7d | 6rz6 |
| ESMFold | **2.3** | 1.7 | **4.0** | 2.8 | 4.5 | 5.9 | **3.7** | 8.6 | 2.7 |
| ColabFold | 3.0 | **1.3** | 4.2 | **1.7** | **2.8** | **3.8** | 5.8 | **1.2** | **2.1** |

Since the available number of membrane proteins in our testset is small, this study does not allow us to give definitive answers on the performance of our model on these types of proteins.

### F.4 CONFIDENCE MODEL EVALUATION

To determine the effectiveness of the confidence model, we have compared how the impact of the number of generated samples on the quality. When having a strong confidence model, the performance with more samples will be monotonically increasing. This analysis is illustrated in Figure 8 for RMSD, SC-RMSD, and for crystal and ESMFold structures respectively. However, if the model only produced very similar poses, then the number of generative samples would not be indicative of the quality of the confidence model. To further investigate the performance of the confidence model, we compare the selective accuracy. For this, we rank the confidence of all top-1 predictions and discard the lowest-ranking ones (according to the confidence model). How this selection compares to an oracle with perfect selection gives insight into the quality of the confidence model. This is shown in Figure 9, where we see that the confidence model works especially well for the RMSD, and is less accurate for the SC-RSMD. In all cases, a higher confidence correlates with a better pose.

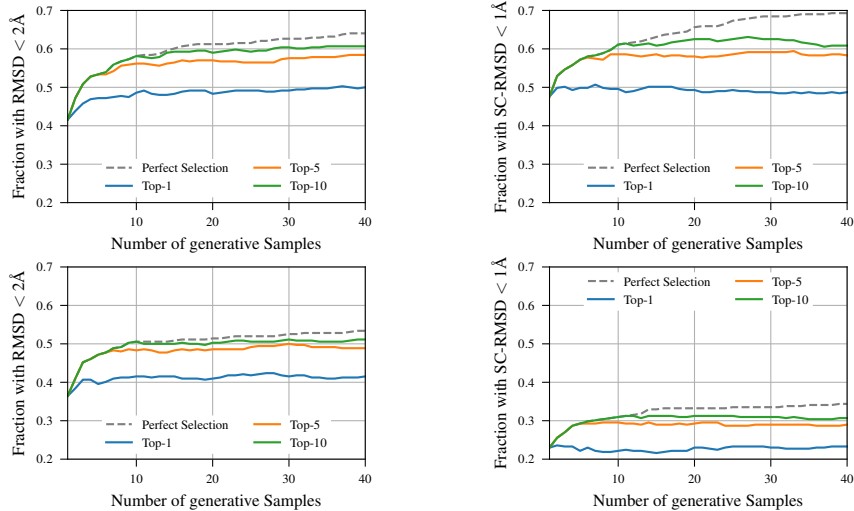

Figure 8: **Performance based on number of generative samples.** Compare the top-1, top-5, and top-10 accuracy based on the number of samples generated by our procedure. In *left*, the RMSD of the ligand can be seen, whereas *right*, the sidechain RMSD is illustrated. In the *top* row, the input are crystal structures, while the *bottom* row uses structures generated by ESMFold.

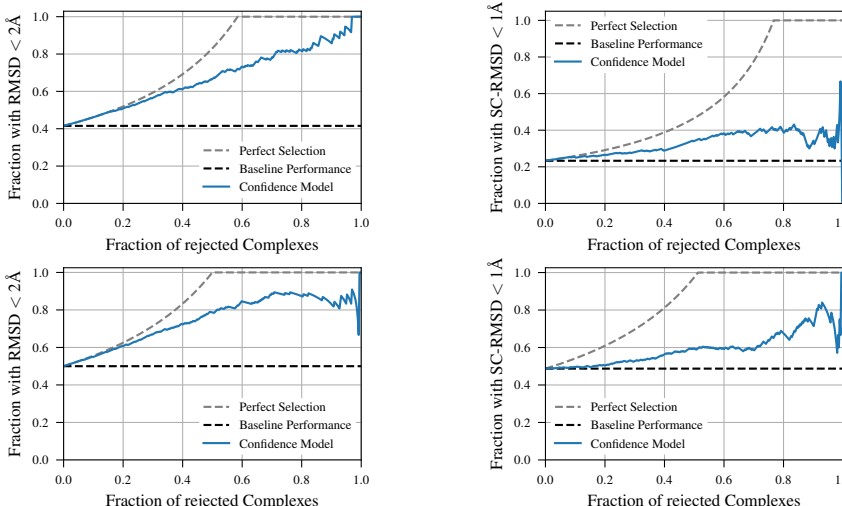

Figure 9: **Selective accuracy of the score-model.** Compare the performance of the model with respect to the confidence model, and a perfect selection. In *left*, the RMSD of the ligand can be seen, whereas *right*, the sidechain RMSD is illustrated. In the *top* row, the input are crystal structures, while the *bottom* row uses structures generated by ESMFold.

### F.5    PERFORMANCE BASED ON QUALITY OF COMPUTATIONAL STRUCTURES

While we saw that the docking results between ESMFold and ColabFold structures did not change much, we will investigate whether the quality of the computationally generated structures impacts the performance. Figure 10 shows the overall quality of the predictions by illustrating the RMSD to the ground truth protein structure in the pocket. We see that more than half of the predictions have an RMSD of $< 2$Å to the ground truth structure. Figure 11 shows the percentage of complexes with a good RMSD and SC-RMSD respectively. For this, we have split the test set into roughly three equally sized parts based on the RMSD of all atoms in the pocket between ESMFold structures and the ground truth crystal structures. We can clearly see that the performance degrades with worse predictions. For structures that are not accurate, our method is not notably better than others. Especially for the sidechains, the prediction quality of our model strongly depends on the quality of the computationally generated structure.

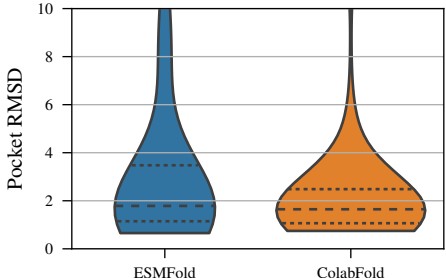

Figure 10: **Pocket RMSD between apo and holo structures.** Apo ESMFold and ColabFold structures have been aligned with the holo crystal structures such that the RMSD in the pocket is the lowest. This figure shows the RMSD of the pocket for proteins in the test set. The dashed lines represent the 25%, 50%, and 75% percentiles respectively. This figure does not show outliers having an RMSD larger than 10Å.

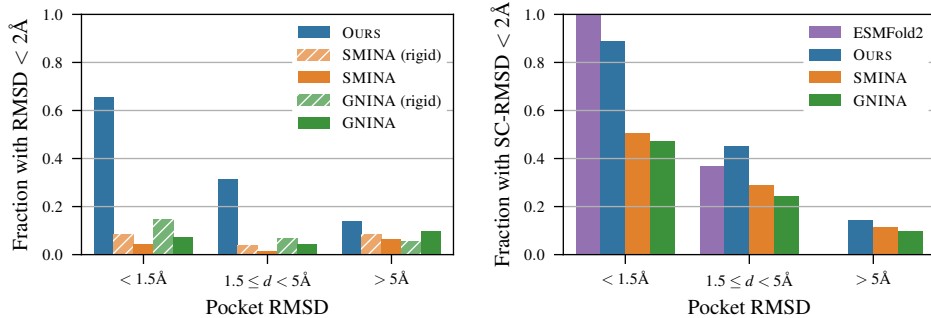

Figure 11: **Model accuracy based on quality of ESMFold predictions.** Comparison of the model accuracy with three different levels of the quality of ESMFold predictions. The predicted ligand (*left*) and sidechain quality (*right*) are evaluated respectively.

### F.6 NUMBER OF REVERSE DIFFUSION STEPS

We evaluated multiple values for the concrete number of reverse diffusion steps on the validation set to determine the best number at inference time. The results are visualized in Figure 12. 30 reverse diffusion steps yielded the best results while not impacting the performance too much. We can see that we could reduce the number of reverse diffusion steps to 20 without losing too much performance. This reduction in reverse diffusion steps could reduce the runtime by up to 33%.

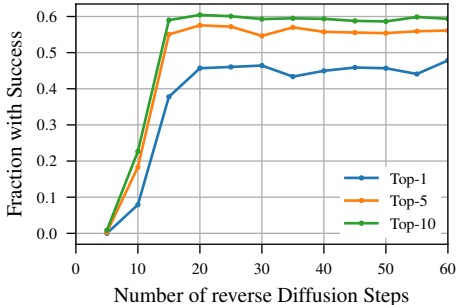

Figure 12: **Comparison of the number of reverse diffusion steps.** Results of the inference with different reverse diffusion steps on the validation set. The values on the y-axis shows the fraction of samples where the RMSD is $< 2$Å and the SC-RMSD is $< 1$Å.

### F.7 IMPACT OF POCKETS FOR CROSS-DOCKING

When comparing works that use site-specific docking, it is important to compare which pockets they used and if the definitions are similar enough not to skew the results. More accurate pockets typically result in better predictions. In Table 16, we see how different pockets influence the results of the performance of our model in the cross-docking benchmark. For this testset, we present the numbers for three different choices of pockets.

1. Use the pocket center definition as we did in training which is defined as the mean $\alpha$-carbon atoms that are within 5Å of any ligand atom. This requires the ground truth ligand and would thus be an unfair comparison. Marked with a *.

2. Use the pocket center definition as Brocidiacono et al. [2023] where they rely on information from multiple ligands [Brocidiacono et al., 2022]. This can be very different from our definitions. Marked with a †.

3. Pre-process the pockets from Brocidiacono et al. [2023] by computing the mean of the $\alpha$-carbon atoms in the pocket. This does not use any additional data and follows a more similar definition to our pocket. These numbers were presented in the main paper.

If the pockets were constructed the same way as in training (i.e., no distribution shift but different data than competitors), we would achieve results improving on the state-of-the-art in all $< 2$Å accuracy metrics. Even giving better predictions than GNINA. When using the exact pockets specified by Brocidiacono et al. [2023], the results are slightly worse than those presented in the paper's main text but still show the same trend.

Table 16: **Cross-docking performance on CrossDocked 2020 with different pockets.** In this table, we present additional results for the cross-docking benchmarks when using different pockets. The method highlighted with * follows our pocket definition presented with access to the ground truth data to compute the pockets as in training. For the results marked with a †, we use identical pocket centers as presented in Brocidiacono et al. [2023].

| | Top-1 RMSD | | Average |
| Method | %<2 | %<5 | Runtime (s) |
| --- | --- | --- | --- |
| DIFFDOCK-POCKET* (10) | **32.7 (31.8)** | **68.2 (71.5)** | **20.6** |
| DIFFDOCK-POCKET† (10) | 26.8 (17.0) | 67.2 (50.5) | 21.4 |
| DIFFDOCK-POCKET† (40) | 28.3 (18.2) | **68.2** (49.6) | 71.6 |

## G    VISUALIZATION OF DOCKING RESULTS

We present the visualization for four different dockings in Figure 13. An animation of the docking process for multiple complexes can be found in our repository at `https://anonymous.4open.science/r/DiffDock-Pocket-AQ32`.

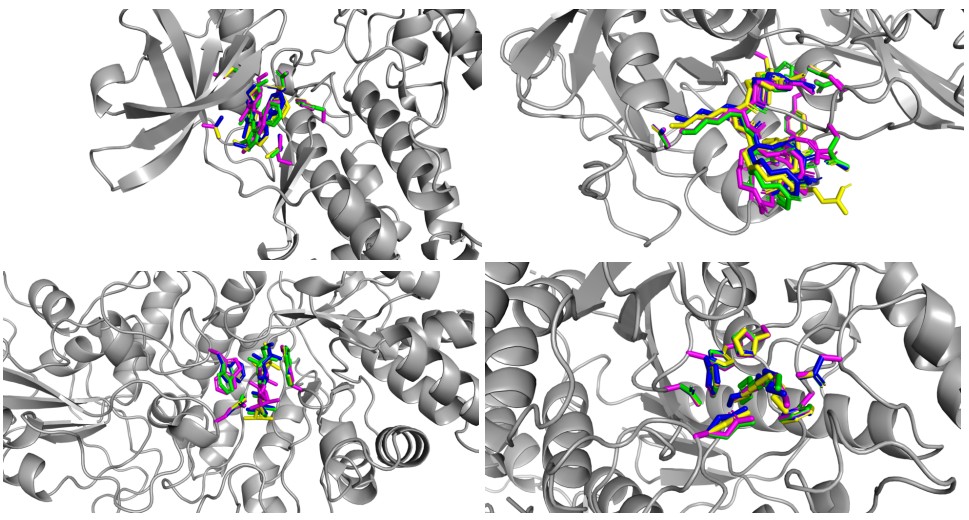

Figure 13: **Flexible docking of unseen complexes.** Visualization of the results of four dockings on arbitrarily selected complexes (*top*: 6a1c, 6hzb, *bottom*: 6md6, 6uii). Four different poses for the sidechains and ligand are presented in different colors.

## H    EVALUATION WITH POSEBUSTERS

We have evaluated our results with the PoseBusters [Buttenschoen et al., 2023] method to determine the percentage of our results which are physically plausible. For this, we used two separate tests implemented by Buttenschoen et al. [2023], one that measures the quality of the predicted complex structures (including intramolecular and intermolecular validity, such as the bond lengths and internal steric clashes in the ligand or its volume overlap with the protein) and the redocking success,

which also takes into account the accuracy of the prediction of the ligand, including that the RMSD between the predicted and true ligand is below 2Å but also checking that the molecules have the same chirality and double bond stereochemical properties. We report these results on our model and baselines for both holo-crystal and ESMFold generated apo structures of the PDBBind benchmark in Table 17.

Table 17: **Results of PoseBusters quality check.** We report the percentage of predictions that pass all Posebuster quality checks required for the docked complex structure and the redocked structure compared to the ground truth ligand.

| Method | ESMFold Structures | | Holo-crystal Structures | |
|---|---|---|---|---|
| | Docking Structure | Re-docking | Docking Structure | Re-docking |
| GNINA (rigid) | 90.4 | 7.0 | **95.8** | **36.3** |
| GNINA | **93.2** | 4.2 | 95.4 | 14.8 |
| DIFFDOCK | 11.4 | 3.6 | 23.4 | 18.4 |
| ESMFOLD | 16.0 | - | - | - |
| DIFFDOCK-POCKET (40) | 21.6 | **10.9** | 29.4 | 17.4 |

The results on holo-crystal structures align with the findings presented by Buttenschoen et al. [2023]: The classical model GNINA outperforms both deep learning models in both the physical plausibility of predicted structures as well as the physical plausibility of *good* predicted structures. Comparing DIFFDOCK-POCKET with DIFFDOCK, we can observe that while the percentage of generated structures that pass all PoseBuster quality checks is higher for DIFFDOCK-POCKET, this advantage disappears when looking at the percentage of structures that are also considered to be a successful redocking attempt and DIFFDOCK even slightly outperforms DIFFDOCK-POCKET.

What we find very promising and something we believe would require further examination is the results on ESMFold structures. In Table 17, we also report that only 16% of ESMFold generated protein structures pass all quality checks when comparing it with the ground-truth ligand. GNINA and DIFFDOCK-POCKET both improve on this number in their generated structures which can be attributed to better sidechain positions. However, although more than 90% of generated structures by GNINA are considered correct, DIFFDOCK-POCKET outperforms all methods when considering successful redocking. This suggests a possible advantage DIFFDOCK-POCKET could have over classical approaches when docking to apo structures, however further examination is needed.

Altogether we can report that on the PDBBind testset DIFFDOCK-POCKET outperforms DIFFDOCK on the percentage of generated structures that pass all PoseBusters checks (with DIFFDOCK being the best deep-learning method reported by Buttenschoen et al. [2023]) and outperforms all considered methods on redocking to ESMFold structures. Further evaluation on the PoseBusters benchmark for both holo-crystals and generated structures as well as comparison to other docking methods is needed to reassure these claims and is an objective of future work.

