# OpenReview forum: "DiffDock-Pocket: Diffusion for Pocket-Level Docking with Sidechain Flexibility"
_ICLR.cc/2024/Conference — Submitted to ICLR 2024_

### Official Review · Reviewer_TENj · 2023-10-27

**Soundness:** 3 good
**Presentation:** 3 good
**Contribution:** 2 fair
**Rating:** 6
**Confidence:** 3

**Summary:**

The paper extends the recently published diffusion-based docking tool DiffDock. DiffDock was originally developed for the challenging task of blind docking. In blind docking the task is to predict the bounded structure of a ligand combined with a protein without additional information as for example potential protein pockets. The following paper extends the idea of DiffDock in the sense of reducing the docking task to a given protein pocket and therefore being able to increase the complexity by considering the flexibility of the amino acid side chains.

**Strengths:**

In my opinion, the paper possesses four key strengths:

1.	The paper extends a state-of-the-art docking method to new functionality by modeling the flexibility of the side chain.
2.	It includes an extensive experimental section and a detailed ablation study in the appendix. I appreciate the authors' transparency regarding a potential bottleneck related to steric clashes when compared to energy-based methods (such as GNINA), as illustrated in Table 5.
3.	The authors have chosen a robust baseline and significantly improved upon it depending on the task. The results obtained using the Apo structures appear quite promising.
4.	Finally, the paper is well-written. The authors present the method, its underlying ideas, and the problem under consideration in a structured manner, along with a comprehensive appendix.

**Weaknesses:**

I find one weakness in the paper: its novelty.

The proposed method, as I understand it (please refer to the 'Questions' section), appears to be primarily a version of DiffDock trained on the protein pocket along with the ligand, rather than the entire holo structure. While I acknowledge that incorporating the flexibility of side chains is a new feature for DiffDock, it's important to note that this concept has already been included in other diffusion-based docking tools. For instance, NeuralPlex, although unfortunately not publicly available.

An intriguing and substantial contribution would have been if the authors had enhanced the PoseBuster benchmark, a topic they discuss in the appendix. From my perspective, it seems this was not the case.

I still have some questions and therefore I am waiting for the rebuttal to give a definitive decision. Although, it is a thoroughly written paper with good results I am hesitant to accept the paper to ICLR due to in my opinion only a slight extension to DiffDock.

**Questions:**

- Do you perform sidechain conformer matching (described in Sec. 3.3) only for the training dataset or as well for the results section, i.e. Table 1 & 2?
- Can you asses or elaborate more on the improvements compared to DiffDock?
  - In Table 1 the improvements on the Apo Structures seem quite significant? Is it purely because you restrict yourself to the pocket?
  - Did you perform significant changes / improvements to the architecture?
-	In Table 2 the results for GNINA and SMINA for the holo structure seem to look quite bad. But isn’t the sidechain for the holo structure already correct?
-	For training the confidence model, you write: “The predictions are then compared with the ground truth training data to assess their quality.” (Sec. 3.4 “Training”). How do you assess the quality? Is it different to how DiffDock performed this task?

---

> ### Author Response · Authors · 2023-11-23
> **Response part 1**
>
> We would like to thank the reviewer for their thorough and detailed review and are happy to address all the suggestions and incorporate them into a revised version of the paper.
> > I find one weakness in the paper: its novelty.
>
> > The proposed method, as I understand it (please refer to the 'Questions' section), appears to be primarily a version of DiffDock trained on the protein pocket along with the ligand, rather than the entire holo structure. While I acknowledge that incorporating the flexibility of side chains is a new feature for DiffDock, it's important to note that this concept has already been included in other diffusion-based docking tools. For instance, NeuralPlex, although unfortunately not publicly available.
>
> Thank you for raising your concerns. In this work, we tried to combine several technical contributions and existing solutions into a clean and working real-world solution addressing the community’s main concerns for deep learning docking. The variety of changes we introduced (see next question) lead to a significant improvement in performance. We see that as a valuable contribution next to novel technical contributions. We tried to highlight the novelty of our work in a global comment.
>
> Regarding your comparison with NeuralPLexer, we would like to point out that although both approaches use diffusion, they operate completely differently. This difference is for both, the prediction of the ligand pose and the prediction of sidechains. NeuralPLexer works with a different training procedure, where they mask out certain parts of the protein and try to reproduce the correct positions. Our approach, however, operates on a reduced product space where we learn the torsional angles of the sidechains. We believe that our approach is a novel alternative.
>
> In addition, we directly tackle the problem of known-pocket docking (large limitation of many existing deep learning docking methods) and provide publicly available code for the community to use and build on.
>
> > Can you asses or elaborate more on the improvements compared to DiffDock?
>
> > Did you perform significant changes / improvements to the architecture?
>
> The two most apparent contributions are the inclusion of the binding pocket in the prediction and the modeling the sidechains close to the binding site as flexible. Introducing the flexibility of the sidechains requires changing the architecture, as the model needs to learn the scores of the torsional angles of the sidechains as well. Compared to the existing blind docking software DiffDock, we additionally introduce a graph to the score model that uses all atoms of the protein, instead of only the C-Alpha atoms. This is only possible due to the introduction of the binding pocket, as this reduces the memory footprint and computational effort needed for operating on all the atoms. This atom-based graph also enables modeling the sidechain atoms. Having this additional information on all the atoms and predicting the sidechain positions is also necessary to learn to avoid steric clashes, which the model would not be able to do otherwise.
>
> To improve the performance on a wider variety of protein conformations, we have introduced split-based training that uses computationally generated ESMFold predictions whenever they are similar to the ground truth crystal structures. In that context, we have introduced sidechain conformer matching that aligns the sidechains of the apo structures such that they can be used efficiently during training. We carefully introduced a penalty term into this process that reduces the number of steric clashes, so that computationally generated (apo) structures can be used in a meaningful way during training.
>
> We further introduced a confidence model that was trained to jointly predict the quality of the sidechains and the ligand pose. This approach was fine-tuned as well on computationally generated structures so that it can better assess the quality of diverse structures. To further enhance the results and to prevent the problem of overdispersion at inference, we introduce a low-temperature sampling approach.
>
> To this end, we demonstrated that the proposed changes allow us to outperform existing approaches for cross-docking and docking to PDBBind in almost all benchmarks. Given the critical importance of these tasks, we believe our work and changes can have a significant impact on a wide variety of biological research.

---

> ### Author Response · Authors · 2023-11-23
> **Response part 2**
>
> > Do you perform sidechain conformer matching (described in Sec. 3.3) only for the training dataset or as well for the results section, i.e. Table 1 & 2?
>
> Thank you for raising these questions. Although we already tried to clarify this in the initial version of the paper, we now see that it still left some room for interpretation and ambiguity. We have reworked section 3.3 to be clearer, but would still like to elaborate on your questions.
>
> In short, we only perform conformer matching during training and only for apo structures. During inference we apply random noise to the sidechains, so performing conformer matching would not provide any helpful information on the sidechain positions for the model. (It could however leak the ground truth crystal test data into the selection of the pocket and flexible sidechains, which is another reason why we do not use it during inference.) The results presented in Table 1 and 2 were performed without sidechain conformer matching. We consider the results one could achieve with sidechain conformer matching as the optimal target. Hence, in Figure 4 we display the best achievable result with conformer matching.
>
> > In Table 1 the improvements on the Apo Structures seem quite significant? Is it purely because you restrict yourself to the pocket?
>
> To answer your question, we have added further evaluation in Appendix F.2 of the revised manuscript which emphasizes the importance and advantage of flexible docking. Although reducing the protein to its pocket already significantly improves docking for ESMfold structures, our results clearly show the benefit of modeling flexibility and the apo/holo split.
>
> > In Table 2 the results for GNINA and SMINA for the holo structure seem to look quite bad. But isn’t the sidechain for the holo structure already correct?
>
> The holo structure is indeed already correct, however, in this case all the methods randomly perturb the angles of the bonds of the flexible sidechains. This means that for the flexible sidechains, no ground truth atom positions are used for the predictions (except the backbone atoms). This functionality was already integrated in SMINA and GNINA and we handled it similarly for our method as to not introduce any data leakage. This explains the poor performance of these methods despite correct input. We clarified this in Section 4 in the revised version of our manuscript.
>
> Perhaps it is also important to mention that although they do not specifically talk about their sidechain prediction accuracy, in the paper where its authors introduce GNINA they themselves report that their sidechain prediction model does not improve the overall accuracy of docking:
> "the difference in ligand RMSD between flexible and rigid docking for the top pose varies widely between systems and there in no clear advantage in flexible docking" [2].
> Which we can also witness in our reported results in Table 1 and 2. Since GNINA does much better on rigid docking, the much worse docking on holo-crystal structures with flexibility can be attributed (together with the more computationally difficult problem of flexibility modeling) to the incorrect estimation of sidechain conformations, which would align with our findings and reported results.
>
> > For training the confidence model, you write: “The predictions are then compared with the ground truth training data to assess their quality.” (Sec. 3.4 “Training”). How do you assess the quality? Is it different to how DiffDock performed this task?
>
> We tried to further clarify this in Appendix C.2. Yes, our confidence model is different because it aims to predict the quality of both the sidechain positions and the ligand pose simultaneously. We train the confidence model by predicting whether the ligand pose is within 2A of the ground truth and the sidechain atoms within 1A. This way, our confidence model approximates the joint probability that both the ligand and sidechain prediction is correct.
>
>
> [1] Buttenschoen, M., et al., "PoseBusters: AI-based docking methods fail to generate physically valid poses or generalise to novel sequences," 2023.
>
> [2] McNutt, A., et al. "GNINA 1.0: Molecular docking with deep learning," in Journal of cheminformatics, vol. 13, no. 1, pp. 1–20, 2021.

---

> > ### Comment · Reviewer_TENj · 2023-12-02
> >
> > I really appreciate the effort the authors put into the rebuttal. Thank you for the additional experiments to understand the method’s capabilities better.
> >
> > I am still concerned about the novelty because of the new experiment in Appendix F.2. It seems that DiffDock alone would also profit from a reduction to only the pocket information.
> > Nevertheless, I will raise my score slightly to support the acceptance of the paper because the authors discuss in detail their method, and I appreciate the extensive appendix material. The results are improved compared to previous methods (as already in the original manuscript). Additionally, I believe the authors addressed all reviewers' comments in detail. Overall, the paper is well-written and contributes to an interesting field of machine-learning-based docking.

---

> ### Comment · Area_Chair_2Xyp · 2023-12-02
> **Does the response address your concerns?**
>
> @Reviewer TENj，
>
> I would appreciate it if you could review the response and adjust your feedback and rating as necessary.
>
> AC

---

### Official Review · Reviewer_Mo9h · 2023-11-01

**Soundness:** 2 fair
**Presentation:** 4 excellent
**Contribution:** 2 fair
**Rating:** 5
**Confidence:** 4

**Summary:**

The paper introduces "DiffDock-Pocket", a diffusion-based algorithm tailored for molecular docking, emphasizing the prediction of ligand poses within specific protein binding pockets. It also incorporates receptor flexibility and sidechain positioning near the binding site. The model showcases state-of-the-art performance on the PDBBind benchmark, with exceptional results when applied to in-silico generated structures.

**Strengths:**

1. The proposed method builds upon DiffDock, incorporating sidechain prediction, which results in superior performance in site-specific docking compared to other existing methods.
2. The paper is well written and follows a clear and logical structure, facilitating understanding and evaluation of the proposed method.
3. The paper provides an anonymous repository with resources to replicate the presented results, which significantly contributes to the advancement of this field of research.

**Weaknesses:**

1. The technical contribution is incremental. The paper mainly extands DiffDock’s idea with sidechain prediction
2. The paper claims that previous models like DiffDock only consider rigid receptor. However, this might be misleading as previous models incorporate just residue-level features, which do not utilize, and thus, do not need to incoorporate sidechain. The primary distinction is that DiffDock-Pocket extends DiffDock by adding sidechain prediction. Both models do not account for backbone flexibility, which should be clarified.
3. The paper predominantly addresses site-specific docking, where the ground truth pocket is provided. A more comprehensive evaluation, including scenarios where the pocket is predicted by other models, would enhance the practical applicability of the findings.
4. The method for curating the pocket center and selecting sidechains for modeling flexibility (within 3.5Å of any ligand) relies on the ligand's position, which could lead to potential data leakage. While the paper mentions that users will manually set these parameters during inference, further details and implications of this process need clarification.
5. The significant improvement observed with in-silico structures generated by ESMFold2 might be attributed to additional training on this dataset. Further ablation studies are necessary when no in-silico structures are used for training.
6. Previous work [1] has demonstrated that traditional methods like SMINA can achieve rapid inference speeds and high performance in docking when the pocket location is provided. The paper indicates that these traditional methods become significantly slower when sidechain flexibility is considered. A more comprehensive comparison, including performance and speed evaluations of traditional models with and without sidechain flexibility, would provide a more balanced and informative perspective.

[1]. Yu, Yuejiang, et al. "Do deep learning models really outperform traditional approaches in molecular docking?." *arXiv preprint arXiv:2302.07134* (2023).

**Questions:**

1. Is there any ablation study conducted to assess the impact of training on structures predicted by ESMFold?
2. Have there been any experiments conducted on training DiffDock with a given pocket but without considering sidechain flexibility? Such experiments could better highlight the benefits of modeling sidechain flexibility.

---

> ### Author Response · Authors · 2023-11-23
> **Response part 1**
>
> We would like to thank you for your thorough feedback and have tried to address all your concerns in a revised version of the document. We are happy to answer all your remaining questions.
> > The technical contribution is incremental. The paper mainly extands DiffDock’s idea with sidechain prediction
>
> Thank you for raising your concerns about our contributions. In this work, we tried to combine several technical contributions and existing solutions into a clean and working real-world solution addressing the community’s main concerns for deep learning docking. The variety of changes we introduced (all-atom architecture, pocket reduction, sidechain flexibility, training on computationally generated structures, sidechain conformer matching, low-temperature sampling, …) lead to a significant improvement in performance. We see that as a valuable contribution next to novel technical contributions. We tried to highlight the novelty of our work in a global comment.
>
> > The paper claims that previous models like DiffDock only consider rigid receptor. However, this might be misleading as previous models incorporate just residue-level features, which do not utilize, and thus, do not need to incoorporate sidechain. The primary distinction is that DiffDock-Pocket extends DiffDock by adding sidechain prediction. Both models do not account for backbone flexibility, which should be clarified.
>
> Thank you for bringing this to our attention. We agree with your points and have changed the document in various places to tone down and clarify these claims throughout the paper. However, it is important not to underestimate the importance of explicitly predicting the sidechain rearrangement. While it is true that DiffDock can implicitly reason about flexible sidechains the fact that it does not predict the sidechain rearrangement makes its prediction in case of flexible sidechains very hard to use in any downstream application like affinity prediction where accurate atomic structures are necessary.
>
> > The paper predominantly addresses site-specific docking, where the ground truth pocket is provided. A more comprehensive evaluation, including scenarios where the pocket is predicted by other models, would enhance the practical applicability of the findings.
>
> Although not on the PDBBind dataset, but for cross-docking, we have evaluated a variety of different pocket definitions for our approach in Appendix F.7, where we demonstrate good performance even with out-of-distribution pocket definitions. Furthermore, we believe that our model could be fine-tuned to operate on pockets predicted by a certain approach to create an end-to-end approach.
>
> > The method for curating the pocket center and selecting sidechains for modeling flexibility (within 3.5Å of any ligand) relies on the ligand's position, which could lead to potential data leakage. While the paper mentions that users will manually set these parameters during inference, further details and implications of this process need clarification.
>
> We believe our pocket definition to be robust, as it is defined by the mean of the C-Alpha atoms close to the binding site instead of relying on the ligand atoms. Similarly, the selection of the flexible sidechains has very little potential for data leakage. The way we handle the flexible sidechains is by first obscuring the sidechain positions with random noise and then creating the graphs and edges with this obscured information. This way, the model is not aware which sidechains are flexible, reducing the possibility of data leakage. We appreciate this comment, and have clarified this in Appendix C.4.

---

> > ### Author Response · Authors · 2023-11-23
> > **Response part 2**
> >
> > > The significant improvement observed with in-silico structures generated by ESMFold2 might be attributed to additional training on this dataset. Further ablation studies are necessary when no in-silico structures are used for training.
> >
> > > Is there any ablation study conducted to assess the impact of training on structures predicted by ESMFold?
> >
> > > Have there been any experiments conducted on training DiffDock with a given pocket but without considering sidechain flexibility? Such experiments could better highlight the benefits of modeling sidechain flexibility.
> >
> > We have added Appendix F.2 to address these concerns, where we compare the performance of different models with and without flexibility, and hence with and without training on ESMFold structures. We can see that the rigid model has significantly lower performance on apo structures compared to a model that was trained with flexibility and the ESMFold structures.
> >
> > The results reported in Appendix E, where we evaluated the performance on ColabFold, suggest that the benefit of training on ESMFold structures translates to other in-silico or experimental datasets even when our model was not trained on them.  Because of this, we argue that although training our model on ESMFold structures does attribute to the improvement of the inference results on ESMFold structures, it in general improves the success of docking on proteins with sidechain structures that differ from holo-crystal structures, which is our main goal.
> >
> > > Previous work [1] has demonstrated that traditional methods like SMINA can achieve rapid inference speeds and high performance in docking when the pocket location is provided. The paper indicates that these traditional methods become significantly slower when sidechain flexibility is considered. A more comprehensive comparison, including performance and speed evaluations of traditional models with and without sidechain flexibility, would provide a more balanced and informative perspective.
> >
> > In Table 1, we have reported the average runtime of the different approaches. For GNINA and SMINA, we have listed the runtime of flexible and rigid predictions, as well as the impact on the accuracy. For all SMINA and GNINA runs, the pocket location has been provided. These results show that the performance with sidechain flexibility is about 6-7 times slower than for rigid predictions.
> >
> > If you believe that there are any further concrete evaluations that would be beneficial for the community, we would be happy to incorporate them into a revision of the document

---

> > > ### Comment · Reviewer_Mo9h · 2023-12-03
> > >
> > > I really appreciate the author's effort in rebuttal. The paper is clearly written and contains detailed experiements. But I  remain concerned about the distinctiveness of the paper's contribution, as it's mainly an extansion of the existing work DiffDock. Given this viewpoint, I have decided to uphold my initial assessment and rating.

---

> ### Comment · Area_Chair_2Xyp · 2023-12-02
> **Does the response address your concerns?**
>
> @Reviewer Mo9h，
>
> I would appreciate it if you could review the response and adjust your feedback and rating as necessary.
>
>
> AC

---

### Official Review · Reviewer_XNuQ · 2023-11-01

**Soundness:** 3 good
**Presentation:** 3 good
**Contribution:** 2 fair
**Rating:** 6
**Confidence:** 3

**Summary:**

The paper presents a ligand docking model, DiffDock-Pocket, that, given a target receptor pocket, predicts the conformation of a given ligand in that pocket, as well as the conformation of the pocket´s side chains. The model claims state-of-the-art performance in ligand docking, given either the apo or holo form of the receptor. Notably, the performance increase compared to other models is especially pronounced in the apo case.

**Strengths:**

1. The paper is well-written and the methodology is presented nicely.
2. The way of considering the all atom structure of the side chains pertaining to a target pocket in the diffusion process is novel.
5. The results are convincing.

**Weaknesses:**

1. The model assumes prior knowledge of the pocket. Perhaps one might be interested in allosteric binding sites.
2. The details of the differences between the diffusion process of DiffDock-Pocket and DiffDock are not entirely clear.
3. It is not clear on which subsets of the data the model performs well and vice versa. E.g., what about membrane proteins?
4. The way in which sidechain conformer matching step affects the results is not entirely clear.

**Questions:**

1. How much information relevant to the pocket-docking task is lost by discarding amino acids that are too far away from the binding site? One scenario that comes to mind is membrane receptors, where conformational changes on the extracelluar side induced conformational changes on the intercellular side.
2. When performing "sidechain conformer matching", are the bond lengths of the computationally generated structures heterogeneous? Can you elaborate on how the imperfect conformer matching affects the results?
3. Does the diffussion on the product space come with the requirement that the torsional updates don't induce angular velocity, as in DiffDock?
4. How do the receptor's atom and amino acid level graphs tie into eachother?
5. In Table 1, there is clearly a performance increase with DiffDock-Pocket, particularly in the apo protein case. When comparing the apo and holo form of examine proteins, what is the proportion of conformational changes due to backbone flexibility and conformational changes due to side chain flexibility? How would the model perform if the majority of the conformational changes were in the backbone?
6. Would it be possible to use DiffDock to identify the pocket and then DiffDock-Pocket to refine the prediction, in an end-to-end fashion?

---

> ### Author Response · Authors · 2023-11-23
> **Response part 1**
>
> We appreciate the effort and knowledge that went into writing this review and are happy to address all remaining questions and concerns.
>
> > The model assumes prior knowledge of the pocket. Perhaps one might be interested in allosteric binding sites.
>
> > How much information relevant to the pocket-docking task is lost by discarding amino acids that are too far away from the binding site? One scenario that comes to mind is membrane receptors, where conformational changes on the extracelluar side induced conformational changes on the intercellular side.
>
> With regard to docking results on the benchmarked datasets, we can report that having information on the whole protein, as opposed to the reduced pocket, does not significantly influence predictions on average. As we describe in section 3.1, pocket knowledge can also be introduced by centering the ligand’s initial random configuration around the pocket’s center without removing any amino acids. We have evaluated this method and compared it to the pocket reduction method on holo-crystal and ESMFold structures and, in both cases, found that the pocket reduction method does not lose accuracy. In some cases, it even slightly increases the percentage of ligand structure predictions under the threshold of 2 Angstroms RMSD. However, by removing parts of the protein, the model performs significantly faster. We used this to increase the size of the model with the caveat that, in some rare instances, the model might make worse predictions but can achieve better accuracy overall. We thank you for the feedback, and have updated the document to highlight the retained performance.
>
> Regarding conformation changes outside of our pocket, it is possible to manually set which amino acids are retained and which amino acids are modeled flexible. Nevertheless, we anticipate that our current model may not perform optimally with choices that are different from the training data (i.e., by modeling amino acids flexible that are far from the ligand). It would, however, be a very useful direction to explore when training a model, perhaps by allowing a user to select two regions for flexible modeling or by randomly picking which amino acids are modeled flexible. We thank the reviewer for bringing it to our attention and are considering a revision that was trained in such a way.
>
> > It is not clear on which subsets of the data the model performs well and vice versa. E.g., what about membrane proteins?
>
> Thank you for bringing this to our attention. We believe this to be a strong addition to our paper and have added a table comparing the docking performance for membrane proteins to Appendix F.3. While the number of membrane proteins in our test set is limited, we still believe that it can give insight into the model’s performance on these types of proteins.
>
> > The details of the differences between the diffusion process of DiffDock-Pocket and DiffDock are not entirely clear.
>
> Thank you for pointing this out. We tried to present the underlying algorithm for training and inference (i.e., diffusion) in a similar fashion as the authors of DiffDock did, so that it can be compared. Please compare Algorithm 2 and 3 in our paper with Algorithm 1 and 2 in DiffDock. As for the diffusion process itself, we introduce a new term $\delta$ that depends on the sidechains. The aim is to approximate the conformer matched sidechains in the loss.
>
> > The way in which sidechain conformer matching step affects the results is not entirely clear.
>
> > When performing "sidechain conformer matching", are the bond lengths of the computationally generated structures heterogeneous? Can you elaborate on how the imperfect conformer matching affects the results?
>
> Thank you for raising these questions. Although we already tried to clarify these points in the initial version of the paper, we now see that it still left some room for interpretation and ambiguity. We have reworked section 3.3 to be clearer and also address these concerns/questions, but would still like to elaborate on your questions.
>
> Sidechain conformer matching is a necessity to use computationally generated structures for training, while still learning correct torsional angles. The bond lengths between the generated structures and the holo crystal structures can differ, and therefore it is not possible to perfectly match the holo structure by only torsional rotations. The different bond lengths can occur because of inaccuracies of in-silico modeling, as well as chemical changes the bonds undergo during binding. Hence, when we would simply apply the same angles that the holo structure has onto the in-silico/apo structure, we would not necessarily produce the conformation that aligns closest to the holo-structure. To minimize this issue, we only use computationally generated structures during training that are similar enough (in terms of RMSD) to the ground truth holo crystal structures.

---

> > ### Author Response · Authors · 2023-11-23
> > **Response part 2**
> >
> > > Does the diffussion on the product space come with the requirement that the torsional updates don't induce angular velocity, as in DiffDock?
> >
> > The reason for the introduction of torsional updates that induce no angular velocity in DiffDock is the need to disentangle the torsional from the roto-translational degrees of freedom. In our case, we keep this convention for the disentanglement of the degrees of freedom of the ligand. When it comes to defining the direction of update of the torsions of the sidechains of protein, we always rotate the side that does not contain the protein backbone. This simple convention makes the update of the sidechains conformation disentangled from the rototranslation of the ligand w.r.t. the protein without requiring any additional Kabsch alignment. We note that in practice this is very similar to the induction of no linear or angular velocity in the protein due to the significantly larger size of the rest of the protein compared to the individual sidechain. We clarified this point in the manuscript in Appendix C.3.
> >
> > > How do the receptor's atom and amino acid level graphs tie into eachother?
> >
> > We have improved and extended the description of our architecture in Appendix C.1 to reflect the methodology better. As for a concrete answer to your question:
> > We have three graphs where the nodes describe 1) ligand atoms, 2) protein atoms, and 3) residues, respectively.
> > Inter-graph edges are formed based on a k-nn graph and the closest neighbors of the atom positions (or bonds for the ligand graph). For the residue-level graph, we use the position of C-Alpha atoms.
> > Similarly, we form edges between all pairs of graphs. For the graphs with the ligand, namely ligand-atom, ligand-residue, we again use the closest neighbors of the atom positions and C-alpha atoms for the residue-level graph. For the atom-residue graph, we connect each atom to the residue to which it belongs.
> >
> > The edges of the ligand-atom, and ligand-residue, have to be built for each step in the reverse diffusion process, where we update the atom positions of the flexible sidechains and ligand atoms. During the forward pass of the model, messages are passed between these nodes and edges by applying message passing with torsional convolutions. With this architecture, all graphs are connected and contribute to the predictions.
> >
> > > In Table 1, there is clearly a performance increase with DiffDock-Pocket, particularly in the apo protein case. When comparing the apo and holo form of examine proteins, what is the proportion of conformational changes due to backbone flexibility and conformational changes due to side chain flexibility? How would the model perform if the majority of the conformational changes were in the backbone?
> >
> > We decided to model only the sidechain positions flexible, because of findings in previous work that suggest most changes occur in these atoms during docking [1]. However, for the apo case, we agree that the quality of the computationally generated structure has a significant impact on the docking results and sidechain predictions.
> >
> > We improved Appendix F.5 in this revision of the document, where we investigated this impact. Generally the difference in sidechain RMSD correlates with the changes in the backbone, as a small change in the backbone accounts for a much larger difference in sc-RMSD (since it affects the positions of the connected sidechain atoms as well) than the same amount of change in a single sidechain atom. In these studies, we can see that the performance of our model significantly decreases when the difference between apo and holo structures is large. As for the ligand docking, our model performs better than other methods even for a sidechain RMSD between 1.5 and 5A. However, as you noted, the sidechain prediction quality more heavily depends on the quality. For an RMSD of 1.5 and 5A, there is no advantage over other approaches. We can attribute this to conformational changes in the backbone.
> >
> > Side note: During training, we combated the problem of difference in the backbone by only relying on apo structures that are similar enough to the holo structures (i.e., have a sufficiently low aligned RMSD). Hence, the model should not learn to compensate for a change in the backbone with unrealistic predictions for sidechains (e.g., by predicting sidechains far away to minimize the distance to the backbone).

---

> ### Author Response · Authors · 2023-11-23
> **Response part 3**
>
> > Would it be possible to use DiffDock to identify the pocket and then DiffDock-Pocket to refine the prediction, in an end-to-end fashion?
>
> We see two different ways on how predictions in an end-to-end fashion could be achieved with DiffDock. Firstly, one could construct a joint training of both models, where in the first step the architecture of DiffDock predicts the ligand pose, and then this is used as a starting position for DiffDock-Pocket. Both models could then be optimized jointly, with the potential to improve blind docking albeit a longer runtime. However, due to the long training time such a procedure would take, it is not possible to address this within the rebuttal period.
>
> The second way would include using both models with the currently trained models in sequence. First, DiffDock predicts a ligand pose, which is then used as an input to DiffDock-Pocket. Here, one has to consider a suitable center for the pocket. Since DiffDock can predict ligands in multiple pockets, we can either only use the top-1 prediction, or use multiple samples from DiffDock to first identify the pockets. Multiple ligand positions in the pocket could then be averaged to produce a robust prediction for the center. As DiffDock can predict a ligand pose within 5A in 63.2% of cases [2, Table 5], it can be meaningful to further refine the results with DiffDock-Pocket.
>
> We believe that both approaches are interesting and thank the reviewer for bringing this to our attention and would like to explore this further.
>
> [1] Jordan J. Clark, et al., "Inherent versus induced protein flexibility: Comparisons within and between apo and holo structures," in PLOS Computational Biology, vol. 15, no. 1, pp. e1006705, 2019.
>
> [2] Gabriele Corso, et al., "DiffDock: Diffusion Steps, Twists, and Turns for Molecular Docking," in International Conference on Learning Representations, 2023.

---

### Official Review · Reviewer_rxhe · 2023-11-01

**Soundness:** 3 good
**Presentation:** 3 good
**Contribution:** 2 fair
**Rating:** 3
**Confidence:** 4

**Summary:**

This article establishes a diffusion model to generate the binding pocket of proteins. The author demonstrates that the structure generated by DIFFDOCK-POCKET outperforms other approaches.

**Strengths:**

The structure of the binding pocket is difficult to predict due to multiple conformations, the absence of complex PDB data, etc. The diffusion model established in this article may provide some insights into solving the problem of predicting the structure of binding pockets bound with ligands.

**Weaknesses:**

1. The authors do not present the performance of the model on specific proteins but provide only the overall performance. In other words, the authors should provide 2-3 examples comparing the 3D structure (atom level) of the pocket with the PDB structure.
2. In the introduction, “The 3D structure of each protein governs the possible interaction partners and, consequently, determines its function.” Not all proteins bind other molecules and then determine their functions, such as antibodies, IDP, structural proteins like 1POK, etc.
3. The performance is only compared with computational methods (SMINA/GNINA). Other deep learning methods are missed.

**Questions:**

1. In the abstract, “When a small molecule binds to a protein, the 3D structure of the protein and its function change.” Why will the function of protein change when it binds molecules? The function of a protein is decided by its sequence and structure.
2. In the results section, the authors only use RMSD to assess the performance of the models. Why don't they use other standards at the same time, such as TM-score, to evaluate the performance?
3. The authors only use an RMSD below 2 angstroms in Table 1 and 1 angstroms in Table as a binary criterion. What makes this discrepancy?  What are the successful rates for different methods at 1/1.5/2 angstroms?
4. In the introduction, “When a molecule (ligand) interacts with a protein (receptor) and binds to it, they form a new complex with a different 3D structure and function.” How to quantitively define the “new” and “different”?
5. What about the performance of VINA for docking prediction in PDBBind docking?  Also why VINA's performance is significantly worse than other methods in Table 3? It is generally believed the best method in terms of binding prediction.

---

> ### Author Response · Authors · 2023-11-23
> **Response part 1**
>
> Thank you very much for your specific and thorough feedback that helps us to improve our work. We responded to all your questions below and have addressed all your concerns in our paper.
>
> > The authors do not present the performance of the model on specific proteins but provide only the overall performance. In other words, the authors should provide 2-3 examples comparing the 3D structure (atom level) of the pocket with the PDB structure.
>
> We believe this to be an excellent illustration of our method and have added visualizations of the reverse diffusion process for various proteins to the [GitHub repository]((https://anonymous.4open.science/r/DiffDock-Pocket-AQ32/visualizations/README.md)) and renderings of the final poses to Appendix G.
>
> Additionally, we also evaluated our model on membrane proteins in Appendix F.3. We believe that this further addresses your concerns about the model’s performance on specific (types of) proteins.
>
> > In the introduction, “The 3D structure of each protein governs the possible interaction partners and, consequently, determines its function.” Not all proteins bind other molecules and then determine their functions, such as antibodies, IDP, structural proteins like 1POK, etc.
>
> > In the abstract, “When a small molecule binds to a protein, the 3D structure of the protein and its function change.” Why will the function of protein change when it binds molecules? The function of a protein is decided by its sequence and structure.
>
> > In the introduction, “When a molecule (ligand) interacts with a protein (receptor) and binds to it, they form a new complex with a different 3D structure and function.” How to quantitively define the “new” and “different”?
>
> Thank you for pointing out the inaccuracy in our wording. We addressed all the places in the paper, clarified the used expressions, and made it more specific. While our prior wording was inaccurate, it was motivated by our objective to emphasize the significant role that molecular binding can play in enabling or inhibiting a protein to fulfill its function(s). Our goal to emphasize this was especially driven by our method's intended application in drug discovery efforts. We hope that the revision makes this clearer while being more accurate.
>
> > The performance is only compared with computational methods (SMINA/GNINA). Other deep learning methods are missed.
>
> As we reported in our related work review, deep learning methods generally do not account for binding sites or flexibility. NeuralPLexer [1], a recent deep learning model that does predict receptor atoms, does not provide available code, thus making evaluation infeasible. Similarly, the source code of DockGPT (Generative Protein Transformer for Docking) [2] is not available either. We believe that this is another benefit of our method, as we have made the source code immediately available to the public.
>
> Benchmarking against DiffDock is a viable alternative, not only for showcasing the enhanced accuracy resulting from incorporating pocket knowledge and sidechain flexibility in our model but also because it claims to have nearly doubled the accuracy compared to previous blind docking deep learning models [6] and is considered one of the best deep learning methods for docking [8]. We thank you for your suggestion, and have clarified this in the paper.
>
> If you are aware of any deep learning methods that perform better than SMINA/GNINA/DiffDock on site-specific docking that are publicly available, we would be happy to incorporate them into another revision.

---

> ### Author Response · Authors · 2023-11-23
> **Response part 2**
>
> > In the results section, the authors only use RMSD to assess the performance of the models. Why don't they use other standards at the same time, such as TM-score, to evaluate the performance?
>
> Thank you for raising your concerns. In our instance, the TM-score would not be a suitable metric to evaluate the performance because our approach keeps the backbone rigid and the TM-score is used to compute the distance between the C-alpha atoms of the protein structures. An extension of the TM-score to all atom positions is not straightforward, as it uses normalizing constants that are specific to the backbone. Changing these computations could be confusing to some readers, as the calculated scores would not have the same meaning anymore (i.e., different thresholds for good predictions). We believe that the sidechain RMSD is therefore a more suitable metric to compare semi-flexible approaches, as used by previous work [5].
>
> > The authors only use an RMSD below 2 angstroms in Table 1 and 1 angstroms in Table as a binary criterion. What makes this discrepancy? What are the successful rates for different methods at 1/1.5/2 angstroms?
>
> We selected the percentage of structures with RMSD below 2 as our metric because they were used by prior works to assess the quality of docking procedures [3,4,5]. Specifically, all methods that we compared DiffDock-Pocket against relied on this metric in their respective publications.
>
> Similarly, for sidechain RMSD, we used the metric also showcased by GNINA (RMSD below 1A) [5]. The threshold is in this case lower because the backbone of the protein is fixed, allowing less variance in the RMSD. In our tests, we saw that when applying random torsional angles to the sidechain bonds, the mean RMSD was only 3.5 Angstroms. We thus believe that a lower threshold than for the ligand is a suitable evaluation. We thank you for raising your concerns and have updated this in our paper to reflect the reasoning.
>
> We fully agree with you that a single binary threshold cannot capture the performance of our docking procedure. We have hence reported the cumulative distribution results for sidechain RMSD results in Figure 4 (and Figure 6) and for docking RMSD in Figure 5 in the appendix. These figures display what percentage of predicted structures are below any given threshold. Table 8, 10, and 11 show the percentiles, which can be used for a more comprehensive evaluation as well. In almost all instances, the performance of the methods at different thresholds aligns with the binary criterion, and DiffDock-Pocket outperforms other methods. For holo-crystal structures, we see that GNINA achieves a slightly higher percentage under 1 Angstrom docking RMSD. However altogether this affects less than 20 percent of generated structures.

---

> ### Author Response · Authors · 2023-11-23
> **Response part 3**
>
> > What about the performance of VINA for docking prediction in PDBBind docking? Also why VINA's performance is significantly worse than other methods in Table 3? It is generally believed the best method in terms of binding prediction.
>
> In selecting our baselines, we aimed to benchmark against freely available state-of-the-art models in deep learning and score-based modeling. For selecting our search-based modeling, we considered that the method GNINA has been reported to consistently outperform VINA on redocking and cross-docking tasks when the binding pocket was defined [3]. This can also explain why it performs significantly better than VINA in Table 3 and hence, why we did not include it in our site-specific docking evaluation. Finally, we would like to note that while they are considered different methods, SMINA (and GNINA) is largely based on different recombination of VINA features that were shown to provide a better performance [7]. We have updated the document to better reflect this.
>
> [1] Zhuoran Qiao., et al., "State-specific protein-ligand complex structure prediction with a multi-scale deep generative model," 2023.
>
> [2] M. McPartlon, J. Xu. "Deep Learning for Flexible and Site-Specific Protein Docking and Design" in bioRxiv 2023.
>
> [3] Alhossary, A., et al., "Fast, accurate, and reliable molecular docking with QuickVina 2," in Bioinformatics, vol. 31, no. 13, pp. 2214-2216, 2015.
>
> [4] C. Hassan. "Protein-Ligand Blind Docking Using QuickVina-W With Inter-Process Spatio-Temporal Integration," in Scientific Reports, vol. 7, no. 1, pp. 15451, 2017.
>
> [5] McNutt, A., et al., "GNINA 1.0: Molecular docking with deep learning," in Journal of cheminformatics, vol. 13, no. 1, pp. 1–20, 2021.
>
> [6] Corso, G.,, et al., "DiffDock: Diffusion Steps, Twists, and Turns for Molecular Docking," in International Conference on Learning Representations, 2023.
>
> [7] Koes, David Ryan, Matthew P. Baumgartner, and Carlos J. Camacho. "Lessons learned in empirical scoring with smina from the CSAR 2011 benchmarking exercise." Journal of chemical information and modeling 53.8 (2013): 1893-1904.
>
> [8] Buttenschoen, M., et al., "PoseBusters: AI-based docking methods fail to generate physically valid poses or generalise to novel sequences," 2023.

---

> ### Comment · Area_Chair_2Xyp · 2023-12-02
> **Does the response address your concerns?**
>
> @Reviewer rxhe,
>
> Could you please check the response and update your review comments and rating (if needed) accordingly?
>
> AC

---

### Author Response · Authors · 2023-11-23
**General response**

We would like to thank all the reviewers for their thoughtful reviews, we truly appreciate the time they took to analyze the paper carefully. We respond to each point raised in the individual responses and distill the novel contributions of our simple yet effective pocket-level docking solution here:

In our work we identify current drawbacks of deep learning docking approaches for common applications such as drug discovery and address them to arrive at an effective solution with high potential impact:

1. Incorporating prior knowledge about the binding pocket: In most drug discovery applications the small molecule binding site is already known - we investigate several solutions for leveraging this additional prior knowledge such as changing the sampling distribution of our diffusion process and reducing the search space.

2. Physically plausible structures: DL docking methods have been found to frequently produce unrealistic conformations, including e.g., steric clashes with the protein. To avoid this, we first operate on the all-atom level and navigate the computational difficulty of doing so with TFNs. Second, we introduce a new confidence model training procedure that biases the conformers to avoid steric clashes. To further refine this, we demonstrated the impact of low-temperature sampling to prevent the problem of overdispersion

3. Sidechain conformations are crucial for downstream analyses: In many downstream analyses of docking computations, such as binding affinity calculations, the side-chain conformations are just as important as the ligand conformation. We additionally predict the sidechain conformations with high accuracy for which we introduced a side-chain conformer matching training procedure to alleviate distribution shift between training and inference time.

Thus, we combine several of our technical contributions and existing solutions into a clean and working real-world solution addressing the community’s main concerns for deep learning docking. We see that as a valuable contribution next to novel technical contributions.

---

### Meta-Review · Area_Chair_2Xyp · 2023-12-10

**Metareview:**

The paper presents a diffusion model, DiffDock-Pocket, aimed at predicting ligand poses within specific protein binding pockets.

Reviewer 1 appreciates the potential insights offered by the model, but criticizes the lack of performance presentation on specific proteins and the absence of deep learning methods in the performance comparison. He/she also questions the changes in protein function when molecules bind, and the criteria for successful rates in Table 1 and Table 3.

Reviewer 2 commends the paper's clarity and the novelty of incorporating sidechain prediction. However, He/she notes the model's limitation in assuming prior knowledge of the pocket and question the potential loss of information relevant to the pocket-docking task when discarding amino acids that are too far away from the binding site. He/she also asks about the effect of sidechain conformer matching on the results and whether the diffusion on the product space requires the torsional updates to not induce angular velocity.

Reviewer 3 praises the paper's clarity and the model's efficiency but criticizes its novelty and the potential data leakage due to the method for curating the pocket center and selecting sidechains. He/she suggests the need for more comprehensive evaluation scenarios and questions the impact of training on structures predicted by ESMFold.

Reviewer 4 appreciates the paper's clear structure and detailed experimental section but notes its lack of novelty. He/she suggests the method is primarily a version of DiffDock trained on the protein pocket along with the ligand, and that incorporating the flexibility of side chains is not a new concept. He/she questions whether sidechain conformer matching is performed for the training dataset or the results section, and how the quality of the predictions is assessed during the training of the confidence model.

**Justification For Why Not Higher Score:**

Several of the previously raised issues have been resolved during the discussion period. However, questions regarding the novelty of the paper remain a significant concern.

**Justification For Why Not Lower Score:**

N/A

---

### Decision · Program_Chairs · 2024-01-16

Reject